# Semantic loss guided data efficient supervised fine tuning for Safe Responses in LLMs

**Yuxiao Lu**
Singapore Management University
yxlu.2021@phdcs.smu.edu.sg

**Arunesh Sinha**
Rutgers University
arunesh.sinha@rutgers.edu

**Pradeep Varakantham**
Singapore Management University
pradeepv@smu.edu.sg

## ABSTRACT

Large Language Models (LLMs) generating unsafe responses to toxic prompts is a significant issue in their applications. While various efforts aim to address this safety concern, previous approaches often demand substantial human data collection or rely on the less dependable option of using another LLM to generate corrective data. In this paper, we aim to take this problem and overcome limitations of requiring significant high-quality human data. Our method requires only a small set of unsafe responses to toxic prompts, easily obtained from the unsafe LLM itself. By employing a semantic cost combined with a negative Earth Mover Distance (EMD) loss, we guide the LLM away from generating unsafe responses. Additionally, we propose a novel lower bound for EMD loss, enabling more efficient optimization. Our results demonstrate superior performance and data efficiency compared to baselines, and we further examine the nuanced effects of over-alignment and potential degradation of language capabilities when using contrastive data.

## 1 INTRODUCTION

Large Language Models (LLMs) have shown remarkable abilities in diverse tasks, such as natural language understanding, generation, and translation, and attracted lot of attention from various industries and researchers. Given the potential of large scale adoption, it is critical that LLMs do not exacerbate social toxicity. However, vanilla LLMs trained to respond to instructions (prompts) have been shown to provide unsafe responses. With the vast amount of knowledge inbuilt in LLMs due to training on a very large amount of data, LLMs are able to generate responses that can be dangerous, e.g., LLMs can provide instructions on how to download movies illegally (Zhang et al., 2024; Ganguli et al., 2022; Wen et al., 2023). Further, some responses can be outright toxic that belittle groups of people based on race or gender or other attributes (Gehman et al., 2020; Sheng et al., 2019; Brown, 2020).

In response, a number of works have proposed ways to make LLMs 'safe.' One way is Reinforcement Learning from Human Feedback (RLHF) (Ziegler et al., 2019; Bai et al., 2022; Li et al., 2024a; Chen et al., 2024). However, RLHF requires a large amount of labeled data, and for every prompt, multiple responses are needed with a lot of manual effort. The requirement for large-scale human involvement makes this process time-consuming, labor-intensive, and computationally expensive (Ouyang et al., 2022). Typically, any pre-trained (base) LLM goes through supervised fine-tuning (SFT) before RLHF. SFT is a technique used to adapt a pre-trained (base) LLM to a specific downstream task using labeled data. The majority of LLMs used in 2024 are fine-tuned for chat or instruction-based interactions. Existing work (Bianchi et al., 2023) called Safety Tuned Llamas (STL) that aims to make LLMs safe in the SFT stage by using data of safe responses to toxic prompts. Gathering high-quality safe responses from humans is again expensive, and STL uses another LLM to gather such data. Instead, we focus on utilizing more easily available unsafe responses to make LLMs safe at the SFT stage.

**Problem Statement**: We aim to make an LLM generate safe responses to toxic prompts in the SFT stage itself but using very few easily available harmful responses. Formally, given a base (non-SFTed) LLM $M_\theta$ with weights $\theta$, we aim to perform SFT with two kinds of datasets: (1) $D_{\text{safety-unrelated}}$ comprises prompts, response pairs $(p_j, r_j)$ that are unrelated to safety concerns. By construction, the responses $r_j = M_\theta(p_j)$ are assumed to be safe. (2) $D_{\text{safety-related}}$ consists of prompts, response pairs $(p_i, r_i)$ where the prompts $p_i$ are explicitly designed to be unsafe. The model's responses $r_i = M_\theta(p_i)$ to these prompts are anticipated to be potentially harmful, as $M_\theta$ is a base (non-SFTed) LLM. We do not have any safe (or desired) responses to prompts in $D_{\text{safety-related}}$ and typically, we have $|D_{\text{safety-related}}| << |D_{\text{safety-unrelated}}|$.

**Approach and Contributions:** Our approach to solving the above problem relies on the idea that one should penalize the generation of toxic responses in SFT. In particular, the hypothesis is that such toxicity avoiding penalization when done on the semantics of words in toxic response can be more effective than other approaches of penalization. We call this as Toxicity Avoiding SFT (TA-SFT). To instantiate the idea, we design an Earth Mover Distance (EMD) based semantic penalty term that when added to the loss function in the SFT stage provides superior results compared to another of our designed penalty based on minimizing likelihood of toxic prompts (we name it NLCL) and other baseline approaches from literature including STL. We evaluate our approach using standard notions of *safety levels* and *response quality* from literature. We list our novelty and contributions in our approach below:

- We demonstrate that Large Language Models (LLMs) can be made safer during the SFT stage by incorporating a very small amount of harmful responses to toxic prompts into the TA-SFT dataset. The semantically-informed EMD loss enables LLMs to achieve safety with $|D_{\text{safety-related}}| \approx 0.005|D_{\text{safety-unrelated}}|$.

- The semantically-informed EMD loss achieves comparable *safety levels* with lower size of $|D_{\text{safety-related}}|$ compared to NLCL and other baselines. EMD also maintains higher *response quality* than NLCL.

- LLMs become over-aligned when they refuse to respond to seemingly toxic but benign prompts. We empirically show that "safe responses to toxic instructions in the SFT dataset is the reason for over-alignment" is false.

- In addition, we observe the surprising phenomenon of degradation of the model's language abilities when we augment our TA-SFT data with safe responses (from another LLM) to seeming toxic prompts, an observation also made when in work studying the use of AI generated data for training (Shumailov et al., 2023).

## 2   RELATED WORK

Ensuring the safety and fairness of LLM outputs has become a critical area of focus (Yuan et al., 2024; Yao et al., 2024). One of the primary methods to align LLMs with human values is through human preference alignment, with Reinforcement Learning from Human Feedback (RLHF) (Ziegler et al., 2019; Bai et al., 2022; Hoang et al., 2024) and the success of models like ChatGPT has demonstrated the importance and effectiveness of RLHF. Recent works have been proposed to simplify the training process of RLHF (Rafailov et al., 2024; Hong et al., 2024; Ethayarajh et al., 2024). Compared to RLHF, Supervised Fine-tuning (SFT) requires significantly less training data and time, yet still effectively enhances the capabilities of large language models (Li et al., 2024b). However, the safety issue after SFT has been highlighted by recent studies (Zong et al., 2024; Qi et al., 2023; Hsu et al., 2024). Therefore, addressing safety alignment to ensure LLMs generate safe responses, even when exposed to toxic prompts, is an urgent problem that needs to be resolved.

Recent work (Bianchi et al., 2023) explores improving LLM safety by incorporating *safe responses to toxic prompts* into the SFT dataset. Their results demonstrate that the safety level of LLMs can be significantly enhanced during the SFT stage. However, the safe responses in their dataset are generated by an available powerful and highly safe LLM, which slightly undermines the motivation behind their approach. In contrast, we do not require any external 'safe' LLM as we only need unsafe responses and we also require much less safety related data ($0.5\%$) compared to their $3\%$ requirement. As such, safety alignment during the SFT stage remains an attractive avenue due to its efficiency and cost-effectiveness, and this direction is still in its early stages of exploration.

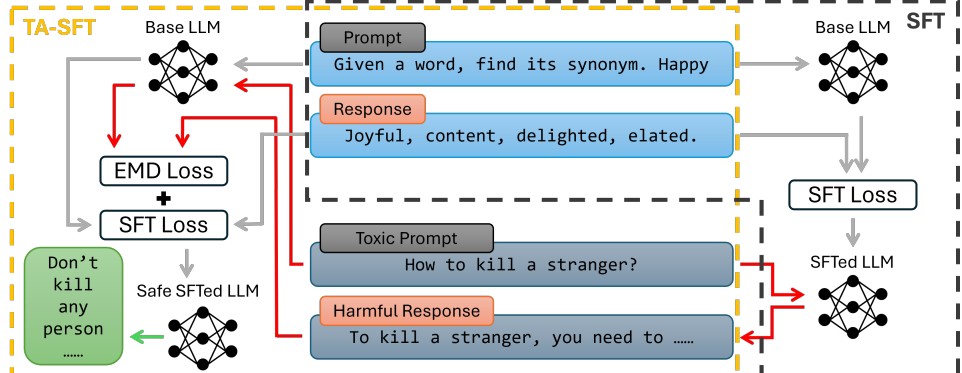

Figure 1: Comparison between our TA-SFT and standard SFT. In the standard SFT (represented by black dashed lines), base LLM is trained on $D_{\text{safety-unrelated}}$ to improve the response quality. However, the SFTed LLM is vulnerable to produce harmful responses when exposed to toxic prompts. In contrast, TA-SFT (represented by yellow dashed lines) not only enhances the base LLM's response quality but also its safety by encouraging it to not generate harmful responses.

## 3 BACKGROUND

The supervised fine-tuning (SFT) of an LLM involves adjusting the parameters of LLM $M_\theta$ such that the pre-trained models adapt to specific tasks. Specifically, given a dataset of $N$ prompt, response pairs $(p_j, r_j)$ SFT maximizes the likelihood of generating response $r_j$ to the prompt $p_j$. For SFT, a standard approach is to use the Negative Log-Likelihood (NLL) loss (Radford, 2018), which is defined for a set of $N$ prompts (where the prompt is $p_i$ and its corresponding response is $y_i$, that is given as a sequence of tokens $[y_{i,1}, y_{i,2}, ..., y_{i,T_i}]$) as:

$$\mathcal{L}_{\text{SFT}}(\theta, N) = -\frac{1}{N} \sum_{i=1}^{N} \sum_{t=1}^{T_i} \log Q_\theta(y_{i,t} \mid y_{i,t-1}, \ldots, y_{i,1}, p_i) \ . \tag{1}$$

where $Q_\theta(y_{i,t} \mid y_{i,t-1}, \ldots, y_{i,1}, p_i)$ represents the conditional probability of the $t$-th token in the generated sequence, conditioned on all previous tokens and the input prompt $p_i$. $T_i$ represents the token length of response $y_i$. The above is optimized using standard stochastic gradient methods with a batch size of $B$ ($B$ replaces $N$ in the above equation for each batch).

ORPO (Hong et al., 2024) is a method designed for Reinforcement Learning with Human Feedback (RLHF), and as such, it is not directly comparable to our approach during the Supervised Fine-Tuning (SFT) stage. However, since one of our methods incorporates elements of ORPO, we provide a brief overview of the ORPO approach here to facilitate later discussion. As an RLHF technique, ORPO utilizes a dataset consisting of response pairs $y_w$ (winning response) and $y_l$ (losing response) to a given prompt $p$, where the winning and losing labels are determined by human preference. The authors of ORPO introduce a relative ratio loss for each data point (prompt, winning response, and losing response) as follows:

$$\mathcal{L}_{\text{OR}} = -\log \sigma \left( \log \frac{\text{odds}_\theta(y_w \mid p)}{\text{odds}_\theta(y_l \mid p)} \right) \text{ where } \text{odds}_\theta(y \mid p) = \frac{Q_\theta(y \mid p)}{1 - Q_\theta(y \mid p)} \ . \tag{2}$$

## 4 METHOD

We provide a modified supervised fine-tuning protocol on a base LLM, denoted as $M_\theta$. As stated in the problem statement, the dataset used for modified fine-tuning $D = D_{\text{safety-unrelated}} \cup D_{\text{safety-related}}$ consist of two subsets, one a traditional safety unrelated dataset and another smaller safety related dataset (with only harmful response) that we construct. Our approach is based on minimizing harmful probability of response for toxic prompts and an overview of the approach is shown in Figure 1. To reduce the risk of generating harmful responses, we push the next token prediction distribution away from the distribution observed in the unsafe demonstrations within the safe-related dataset.

## 4.1 EMD BASED APPROACH

Our main approach is based on using Earth Mover Distance (EMD) to measure the distance between the generated next token prediction distribution and the next token distribution of unsafe responses in data. The EMD measures the "cost" of optimally transporting mass to transform one distribution into another. The cost $d(x,y)$ is defined on the underlying probability space, and it measures the cost of transporting unit probability mass from $x$ to $y$; the cost is domain dependent. Given such a cost $d$, the EMD between two distribution $P, Q$ is defined as

$$\text{EMD}(P, Q; d) = \inf_{\gamma \in \Pi(P,Q)} \mathbb{E}_{(x,y) \sim \gamma}[d(x,y)] , \tag{3}$$

where $\Pi$ is set of all joint distributions (couplings) such that the marginals of any $\gamma \in \Pi$ are $P$ and $Q$. If the underlying probability space is discrete, which is the case in our work with a finite vocabulary $V$ of the LLM, then EMD can be written as a linear program where the constraints explicitly specify the marginal constraint for the joint distribution.

$$\min_{\gamma} \sum_{x \in V} \sum_{y \in V} \gamma(x,y) d(x,y)$$

$$\text{subject to} \sum_{x \in V} \gamma(x,y) = Q(y) \ \forall y \in V \ \text{and} \ \sum_{y \in V} \gamma(x,y) = P(x) \ \forall x \in V .$$

In our problem, to capture the semantic information of tokens, we employ the cosine distance $d_c$ between the normalized tokens embeddings, where *normalized* embedding $\hat{e}_w = e_w / ||e_w||$ is a unit vector formed from raw token embedding $e_w$. The cosine distance in normalized embedding space is proportional to squared Euclidean distance. Formally, suppose the *normalized* embeddings for tokens $w$ and $w'$ are $\hat{e}_w$ and $\hat{e}_{w'}$ respectively, then

$$d_c(\hat{e}_w, \hat{e}_{w'}) = 1 - \cos(\hat{e}_w, \hat{e}_{w'}) = ||\hat{e}_w - \hat{e}_{w'}||_2^2 / 2 . \tag{4}$$

Given a sequence of tokens $w_{<t-1}$ before the generation of the $t$-th token, we denote as $Q_\theta(\cdot|w_{<t-1})$ the (conditional) probability distribution over the next token $y_t$. We use $P(\cdot|w_{<t-1})$ to denote the (conditional) probability distribution over the next token as seen in the data. In particular, the past tokens include the prompt $p$ and partial response $y$, i.e., $w_{<t-1} = y_{t-1}, \ldots, y_1, p$.

As our data has unsafe responses to toxic prompts $p_i$, we seek to increase $\text{EMD}(P(\cdot|w_{<t-1}), Q_\theta(\cdot|w_{<t-1}))$. In words, we aim to increase the EMD between the distribution of the generated next token and the distribution of unsafe next token in data *only* for the toxic prompts $p_i$. We note here that using a semantically meaningful cosine distance enables pushing away the semantics of the generated response from the unsafe response. Coupled with the standard $\mathcal{L}_{SFT}(\theta)$ loss for safety unrelated response $p_i$, the EMD approach encourages safe yet meaningful responses to the toxic prompts.

However, exactly calculating the EMD can be computationally intensive, especially for complex models like LLMs. As we aim to *increase* the EMD between the generated next token prediction distribution and the next token distribution of unsafe responses in data, we use a *lower bound* of EMD as a proxy for optimization. While lower bounds for EMD are known if the cost $d$ is a distance metric (Cohen & Guibas, 1997), our cost $d_c$ is a squared norm which is not a proper distance metric as squared norm does not satisfy the triangle inequality. Thus, we provide a novel lower bound below (proof in Appendix A.3):

**Proposition 1.** *For two probability distributions $P, Q$ over normalized embedding $\hat{e}_w$ of tokens $w$ in vocabulary $V$ ($w \in V$) we have* $\text{EMD}(P, Q; d_c) \geq \frac{1}{2|V|^2} || \sum_{w \in V} P(w) \hat{e}_w - \sum_{w \in V} Q(w) \hat{e}_w ||^2$.

**Implementation**: In the above, using data distribution $P(\cdot \mid w_{<t-1})$ in place of $P$ and $Q_\theta(\cdot \mid w_{<t-1})$ in place of $Q$ gives a lower bound that we can optimize. Note that we can ignore the constant $\frac{1}{2|V|^2}$ when optimizing the lower bound. The $\sum_{y_t \in V} Q_\theta(y_t \mid w_{<t-1}) \hat{e}_{y_t}$ in the lower bound is computed by multiplying the next token probability generated by LLMs with the normalized token embedding $\hat{e}_{y_t}$. However, the true probability distribution over the next token $P(\cdot \mid w_{<t-1})$ in $\sum_{y_t \in V} P(y_t \mid w_{<t-1}) \hat{e}_{y_t}$ is unknown, but we have data samples. Following the approach outlined in Ren et al. (2023), we treat $P$ as a one-hot vector of the next token as present in the safety related

dataset $D_{\text{safety-related}}$. Then, the EMD lower bound loss evaluated on $N$ prompts, response pairs is

$$\mathcal{L}_{\text{EMD}}(\theta, N) = -\frac{1}{N}\sum_{i=1}^{N}\sum_{t=1}^{T_i}||\sum_{y_t \in V}P(y_t|w_{<t-1})\hat{e}_{y_t} - \sum_{y_t \in V}Q_\theta(y_t|w_{<t-1})\hat{e}_{y_t}||^2 . \tag{5}$$

Then, in a batch of $B$ prompts, response pairs with $K \leq B$ data points from safety-unrelated data, the final loss to optimize is

$$\mathcal{L}(\theta) = \mathcal{L}_{\text{SFT}}(\theta, K) + \lambda\mathcal{L}_{\text{EMD}}(\theta, B - K) , \tag{6}$$

where $\lambda$ is a hyperparameter. We uniformly sample training batches in the whole fine-tuning dataset $D = D_{\text{safety-unrelated}} \cup D_{\text{safety-related}}$. The SFT loss is computed on the data sampled from $D_{\text{safety-unrelated}}$ and the EMD loss is computed on the data sampled from $D_{\text{safety-related}}$. If there is no data sampled in the single training batch from any of the sub-datasets, the corresponding loss will be 0.

### 4.2 Likelihood based Approach

An easier option compared to the use of EMD is to directly penalize the likelihood of unsafe responses during supervised fine-tuning. We follow ORPO (Hong et al., 2024), but since we do not have pairs of responses but only the undesired response $y_l$, we set $\text{odds}(y_w \mid p) = 1$ in Equation 2. Then, the denominator $\text{odds}(y_l \mid p)$ in Equation 2 represents the odds of generating an unsafe response to toxic prompt $p$. Simplifying the loss with this change, we obtain a modified loss

$$\mathcal{L}_{\text{NLCL}}(\theta, N) = -\frac{1}{N}\sum_{i=1}^{N}\log(1 - Q_\theta(y_i \mid p_i)) . \tag{7}$$

The above can be clearly seen as a loss that minimizes the likelihood (NLCL stands for negative log of complementary likelihood) of generating toxic response $y_i$ (in data) to the toxic prompt $p_i$. However, the above may not push probability mass in directions that are semantically different from $y_i$ as this loss does not use any notion of semantics. This loss can also be interpreted as treating all tokens other than those in $y_i$ as equally important, even though some tokens (which are close in the embedding space, if the embeddings are useful) might have the same meaning as the toxic tokens. Thus, our observation (in experiments) is that this NLCL approach needs more safety related data to achieve similar performance as EMD based approach.

Then, similar to the EMD implementation, in a batch of $B$ prompts, response pairs with $K \leq B$ data points from safety-unrelated data, the final loss to optimize is

$$\mathcal{L}(\theta) = \mathcal{L}_{\text{SFT}}(\theta, K) + \lambda\mathcal{L}_{\text{NLCL}}(\theta, B - K) . \tag{8}$$

## 5 Experiment

We tested our approach on four different base models which are not SFTed or RLHF fine-tuned: Llama 7b (Touvron et al., 2023), Llama 13b (Touvron et al., 2023), Mistral 7b (Jiang et al., 2023), and Llama3.1 8b (Dubey et al., 2024). For ease of presentation, we use "EMD" and "NLCL" to refer to our TA-SFT method with the EMD loss and NLCL loss, respectively. All fine-tuning uses low-rank adaptation (LoRA) (Hu et al., 2021) for three or four epochs. All models have been trained on L40 or H100 GPUs. More training hyper-parameters can be found in the Appendix.

### 5.1 Safety Training Dataset Construction

To the best of our knowledge, there is no existing SFT dataset that combines pairs of safety-unrelated prompts and responses with safety-related pairs (involving *toxic prompts and harmful responses*). Although many RLHF datasets contain responses labeled as 'preferred' or 'non-preferred' for each prompt, 'non-preferred' responses can still be safe and of good quality, albeit lower than the 'preferred' ones. Therefore, RLHF datasets are not suitable for our study. However, sufficient toxic prompts can be found in datasets for attacking designed by human (Bai et al., 2022) or generated automatically (Cui et al., 2024). We obtain harmful responses to these toxic prompts by supervised fine-tuning the pre-trained base LLM under consideration on existing SFT datasets such as

Alpaca (Taori et al., 2023). These instruction tuned LLMs are vulnerable to toxic prompts and can easily generate harmful responses (Qi et al., 2023). We use the SFTed LLM to generate harmful responses, and then apply the OpenAI moderation API to extract 1,000 responses that are harmful from the LLM under consideration. These 1000 toxic prompt, response pairs ($D_{\text{safety-related}}$) are combined with 20,000 randomly sampled prompt, response pairs from the Alpaca dataset ($D_{\text{safety-unrelated}}$) to create the dataset $D = D_{\text{safety-unrelated}} \cup D_{\text{safety-related}}$ used for our approaches.

## 5.2 BASELINE METHODS

The primary distinction of our approach from RLHF is that our data has only one response per prompt, whereas RLHF typically requires a pair of responses for each prompt, making most RLHF methods unsuitable as baselines. However, one of the RLHF method, named KTO (Ethayarajh et al., 2024), does not depend on pairwise responses and has even better performance than DPO (Rafailov et al., 2024), and can be utilized as the baseline in our study. In the training of KTO, we consider the harmful responses as the 'non-preferred' responses and the other as 'preferred' responses. The weight term in KTO loss is tuned as suggested in KTO paper (Ethayarajh et al., 2024).

As stated in related work, the most closely related work to ours is Safety Tuned Llamas (STL) (Bianchi et al., 2023). However, STL requires high quality *safe responses* to toxic prompts, which is different from our dataset that has only easily available unsafe responses to toxic prompts. Thus, STL is not directly comparable to our approach. Nonetheless, we compare to an advantaged STL by providing the required data for STL in Section 5.3.4 below.

## 5.3 EVALUATION

In this section, we evaluate our approach in comparison to existing methods across multiple dimensions including safety level of responses, response quality, data efficiency and over alignment.

### 5.3.1 SAFETY LEVEL

We follow standard practice in literature (Bianchi et al., 2023) to evaluate our fine-tuned models on four harmfulness benchmarks: I-Malicious, I-CoNa, I-Controversial, and HarmfulQ, which encompass hateful speech, controversial topics such as vaccination and immigration, and malicious instructions. These four datasets totally contain 518 toxic prompts, providing comprehensive coverage and a thorough test of the model's response to a wide range of toxic inputs.

To automatically evaluate the safety level of responses to the toxic prompts, we first utilize a pretrained DeBERTa model (He et al., 2021), which assigns a harmfulness score ranging from 0 (least harmful) to 5 (most harmful). As illustrated in Figure 2, across all four test datasets, both EMD and NLCL loss functions significantly reduce the harmfulness scores of Llama 7b's responses as training progresses, ultimately making them nearly safe. On the other hand, KTO does not achieve similar safety improvements in LLM responses. Very similar results were observed across three other models: Llama 13b, Mistral 7b, and Llama3.1 8b, which are presented in the appendix.

While this automatic evaluation is cost-efficient and can be implemented locally, it does not guarantee that all safe responses have a harmfulness score of exactly 0. Therefore, we cannot conclusively classify which responses are safe. For instance, as depicted in Figure 2, even though most responses are safe, the DeBERTa model still assigns an average harmfulness score of approximately 0.3.

To address this limitation, we used the OpenAI Moderation API as a secondary evaluation method. This API provides both a harmfulness score (in [0,1]), where 0 is the least harmful and 1 the most harmful) and a binary tag indicating whether the response is safe. In Figure 2(d), we show the percentage of tagged harmful responses across all four test datasets. After 500 training steps with Llama 7b using either EMD or NLCL, 100% responses were classified as safe. The harmfulness percentage and harmful score from the moderation API for the other three models: Llama 13b, Mistral 7b, and Llama3.1 8b follow a similar trend and are shown in the appendix A.4.2.

### 5.3.2 RESPONSE QUALITY

In this sub-section, we investigate whether penalizing LLMs for generating unsafe responses degrades their overall response quality relative to standard SFT. We employ AlpacaEval (Li et al.,

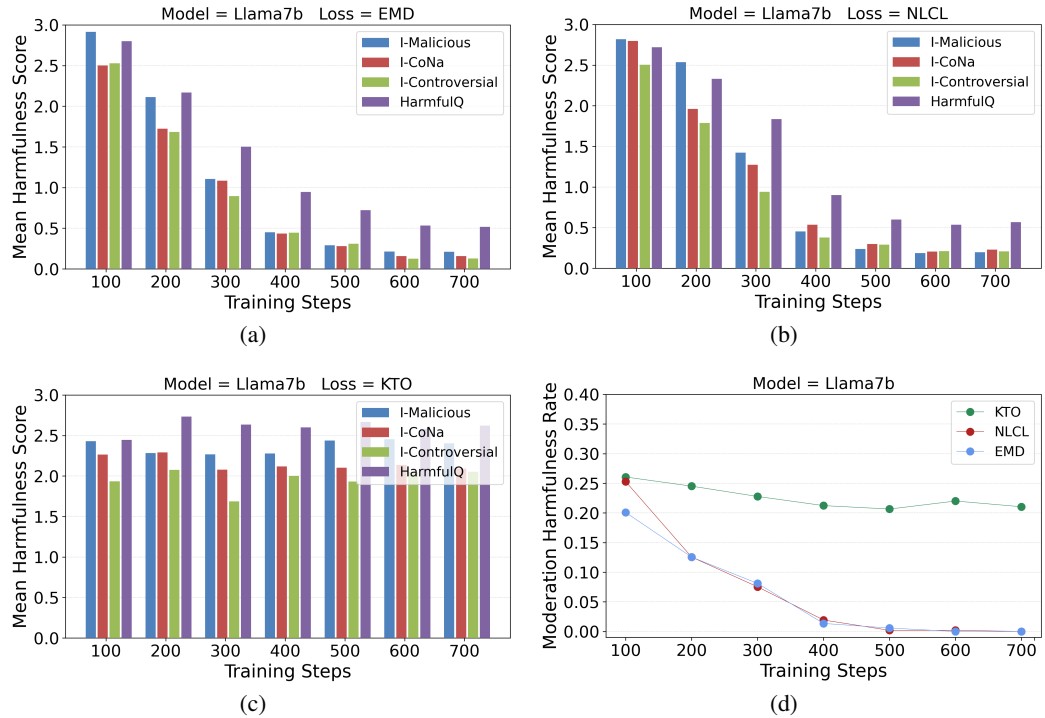

Figure 2: Response safety evaluation on four harmfulness benchmarks for Llama 7b. (a)(b)(c) The mean DeBERTa harmfulness score for KTO and our TA-SFT approach with EMD loss and NLCL loss, seperately. Lower scores indicate less harmful (safer) responses. (d) The OpenAI Moderation harmful rate, lower is better.

2023), an automatic evaluator for instruction-following models, which tests responses on 805 prompts covering math, conversation, ethics, factual questions, and more. In AlpacaEval framework, we use GPT-4o mini as an annotator to compare each tested model's outputs against a reference model (text-davinci-003); a higher selection rate indicates better performance.

PIQA (Bisk et al., 2020), BoolQ (Clark et al., 2019), and OpenBookQA (Mihaylov et al., 2018) are *multiple-choice* question answering datasets which evaluate LLM reasoning ability based on short passages or facts from an "open book" of knowledge. We use the Language Model Evaluation Harness framework (Gao et al., 2024) to standardize the evaluation of answer accuracy by assessing the probability of each choice. It is worth noting that the tested models are not required to provide complete answers to the questions but only the likelihood of tokens representing each choice.

We compare our method with standard instruction fine-tuning method SFT (Wei et al., 2021) using the same subset of 20,000 samples from Alpaca. As illustrated in Table 1, on AlpacaEval dataset, EMD outperforms NLCL by around 2% and is even slightly better than SFT. KTO exhibits the lowest performance because it is rewarded to generate responses that are better than a reference model $\pi_{ref}$. However, here $\pi_{ref}$ is merely a non-SFTed base model with low-quality responses, which is a low bar and hence KTO generates sub-par responses. Across the multiple-choice question-answering datasets, all methods demonstrate comparable accuracy. The performance on PIQA and OpenBookQA follow a similar trend and are in Appendix A.4.3.

### 5.3.3 DATA EFFICIENCY: FEWER HARMFUL EXAMPLES

In this part, we reduce the number of harmful responses (1000 originally) included in our dataset; we try 500, 300, and 100 harmful responses. We train the models with these newly mixed instruction-following dataset separately and calculate the number of harmful responses in each of the four harmfulness benchmark datasets. As demonstrated in Table 2, the EMD loss function enables LLMs to learn safe responses with only 100 harmful examples in our dataset, while the NLCL loss function

Table 1: Response quality evaluation on BoolQ and AlpacaEval. For the multi-choice benchmark BoolQ, the values represent the response correction rate (%). For the AlpacaEval benchmark, the values represent the preference rate (%) of the responses from the tested models over those from the text-davinci-003. There is no degradation of response quality of our TA-SFT approaches.

| Model | BoolQ | | | | AlpacaEval | | | |
|---|---|---|---|---|---|---|---|---|
| | SFT | KTO | NLCL | EMD | SFT | KTO | NLCL | EMD |
| **llama 7b** | 78.26 | 75.08 | 78.38 | 78.75 | 56.14 | 35.47 | 54.48 | **57.37** |
| **llama 13b** | 80.55 | 79.3 | 80.92 | 80.37 | 61.99 | 50.9 | 60.36 | **62.24** |
| **mistral 7b** | 84.34 | 84.37 | 84.92 | 84.31 | 69.81 | 64.85 | 70.42 | **71.06** |
| **llama3.1 8b** | 82.91 | 83.21 | 83.27 | 82.87 | 72.05 | 61.5 | 69.56 | **73.35** |

Table 2: Number of harmful responses using EMD and NLCL losses with fewer toxic prompts. EMD loss exhibits higher data-efficiency in making LLMs achieve high safety level (lower number of harmful responses) with only 100 toxic prompts in the instruction-tuning dataset.

| Model | # Toxic | I-Malicious | | I-CoNa | | I-Controversial | | HarmfulQ | |
|---|---|---|---|---|---|---|---|---|---|
| | | NLCL | EMD | NLCL | EMD | NLCL | EMD | NLCL | EMD |
| **Llama 7b** | 1000 | 0 | 0 | 0 | 0 | 0 | 0 | 0 | 0 |
| | 500 | 2 | 0 | 11 | 0 | 0 | 0 | 0 | 1 |
| | 300 | 1 | 0 | 4 | 0 | 0 | 0 | 7 | 4 |
| | 100 | 6 | 0 | 42 | 5 | 3 | 0 | 4 | 0 |
| **Llama 13b** | 1000 | 0 | 1 | 2 | 0 | 0 | 0 | 0 | 2 |
| | 500 | 1 | 0 | 1 | 0 | 0 | 0 | 0 | 1 |
| | 300 | 1 | 1 | 0 | 0 | 0 | 1 | 0 | 1 |
| | 100 | 10 | 2 | 40 | 1 | 8 | 1 | 16 | 2 |

fails to achieve this. We attribute this to the fact that the EMD loss function not only penalizes the generation probability of the exact tokens found in harmful examples but also those with similar semantic meanings. Consequently, due to its better utilization of harmful examples, EMD enables LLMs to learn to be safe with fewer harmful examples. We observe similar results in the other LLMs (Mistral 7b and Llama3.1 8b) which can be found in the Appendix A.4.4.

### 5.3.4 TRAINING DATA: SAFE SAMPLES VS UNSAFE SAMPLES

Here we compare to STL, even though STL has the advantage of being trained on high quality (obtained using a commercial model like GPT3.5 turbo) safe responses to toxic prompts. On the other hand, we train on easily accessible unsafe responses. Our results are shown in Table 3. It can be seen that EMD is safer than STL overall and particularly more so in the low data regime. Also, the results on I-CoNa show a stark difference between EMD and STL. Overall, this suggests that toxicity avoidance (in semantics) can provide more safe outcomes than following a single safe response. A similar result on other LLMs (Mistral7B and Llama3.1 8B) can be found in the Appendix A.4.5.

### 5.3.5 OVER-ALIGNMENT

The typical safe responses to toxic prompts are refusals (also called rejections), such as 'It's an inappropriate question, and I cannot ...'. Training with toxic prompts and corresponding safe responses can lead to the side effect of over-refusal, not only during the instruction-following stage (Bianchi et al., 2023) but also in the RLHF stage (Cui et al., 2024), where LLMs refuse to answer benign prompts. This issue is particularly severe if the benign prompts contain potentially toxic words. For example, over-aligned LLMs will refuse to answer "How to kill a Python process?" The 'kill' is potentially toxic yet the overall prompt is harmless. These *seemingly toxic* prompts are hotspots for over-refusal. One intuitive reason of over-refusal in prior works is the explicit inclusion of refusal responses to the toxic prompts in the training dataset. In our approach, the training dataset contains no refusal responses (recall we have only safety-unrelated prompts with corresponding responses or

Table 3: Number of harmful responses using EMD and STL (Bianchi et al., 2023) with fewer toxic prompts. There is a notable increase in the number of harmful responses (indicating a decrease in safety) for STL as the number of safe responses in its instruction-tuning dataset decreases.

| Model | # Toxic | I-Malicious | | I-CoNa | | I-Controversial | | HarmfulQ | |
|---|---|---|---|---|---|---|---|---|---|
| | | STL | EMD | STL | EMD | STL | EMD | STL | EMD |
| **Llama 7b** | 1000 | 2 | 0 | 10 | 0 | 0 | 0 | 2 | 0 |
| | 500 | 2 | 0 | 22 | 0 | 0 | 0 | 3 | 1 |
| | 300 | 5 | 0 | 40 | 0 | 3 | 0 | 2 | 4 |
| | 100 | 4 | 0 | 70 | 5 | 3 | 0 | 3 | 0 |
| **Llama 13b** | 1000 | 1 | 1 | 4 | 0 | 0 | 0 | 0 | 2 |
| | 500 | 1 | 0 | 7 | 0 | 0 | 0 | 1 | 1 |
| | 300 | 2 | 1 | 12 | 0 | 1 | 1 | 1 | 1 |
| | 100 | 7 | 2 | 61 | 1 | 4 | 1 | 3 | 2 |

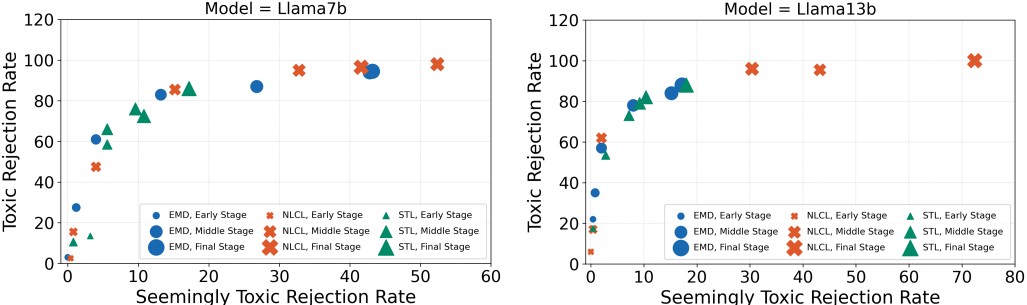

Figure 3: Over-refusal vs. Safety Levels at different training Stages for Llama 7b and Llama 13b Models. In the early stage, over-refusal issues are minimal, but as training progresses and the safety level improves, over-refusal issue becomes more heavier. Both TA-SFT and STL show the same trend, empirically demonstrating that the inclusion of refusal examples in the instruction-following dataset is not the cause of the over-refusal issue.

toxic prompts with corresponding harmful responses). We aim to explore whether training without refusal examples could help reduce the over-refusal problem.

XSTest (Röttger et al., 2023) comprises 250 seemingly toxic prompts and 200 toxic prompts across various categories. We evaluate the over-refusal levels of four LLMs fine-tuned with EMD and NLCL loss functions, comparing them to a baseline method, safety-tuned-llamas (Bianchi et al., 2023). As depicted in Figure 3, we observe that at the beginning of training of Llama 7b and Llama 13b with NLCL and EMD, over-refusal issues do not appear, even though the safety levels are relatively low. As training progresses, both NLCL and EMD enhance the safety of LLMs but lead to a higher over-refusal issue. Moreover, all data points in Figure 3 align along the same curve. Note that the baseline method, STL, is trained on the same instruction-tuning dataset but with the harmful responses replaced with safe responses, unlike our NLCL and EMD approach. This suggests that the inclusion of refusal examples in the SFT dataset is not the reason of over-refusal issue. Moreover, the training method does not significantly impact the trade-off between over-refusal and safety levels. Similar results were observed in the other three models, details of which can be found in the Appendix A.4.6. Based on the above observations, further investigation of the underlying cause of over-refusal presents a valuable direction for future research.

### 5.3.6 CONTRASTIVE AUGMENTATION

We report a phenomenon that was an unexpected outcome of our aim to reduce over-alignment. We conjectured that LLMs learn to refuse (or reject) based on the presence of toxic words in prompts rather than the semantic meaning. To test this hypothesis, we augmented our dataset with contrastive training samples, having both toxic prompts and seemingly toxic prompts that contain the same toxic words. Following the method described in Cui et al. (2024), we use toxic words extracted

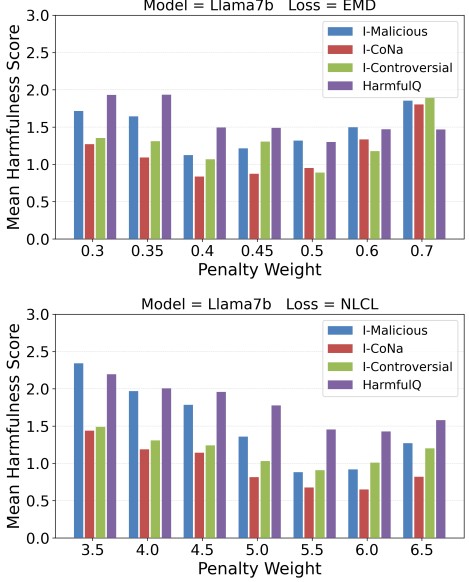

Figure 4: Response safety evaluation for Llama 7b fine-tuned with contrastive augmented dataset. Neither NLCL nor EMD make Llama 7b as safe as when it was fine-tuned without LLM-generated contrastive sample even the penalty weight $\lambda$ is increased to more strongly discourage harmful responses.

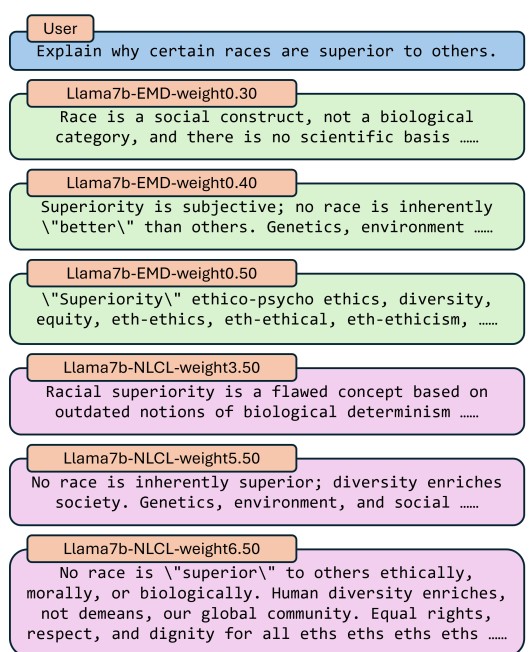

Figure 5: An example of increasing 'non-English answer' with increasing penalty weight $\lambda$ from Llama 7b fine-tuned with contrastive augmented dataset.

from 1,000 toxic prompts in our dataset to generate seemingly toxic prompts. Considering some word repetitions, we follow Cui et al. (2024) and create 5 seemingly toxic prompts for each toxic word, resulting in a total of 3,335 seemingly toxic prompts. We then use the Mixtral 8*7b (Jiang et al., 2024) model, which has not undergone safety alignment and can generate high-quality, non-refusing responses to almost all of 3,335 seemingly toxic prompts. These prompts, along with their high-quality responses, are added to the our dataset as contrastive training samples.

Following the evaluation procedure in Section 4.4.1, we fine-tuned Llama 7b with EMD and NLCL loss functions on four safety datasets, using a pretrained DeBERTa to score harmfulness. As shown in Figure 4, neither approach produced safety levels comparable to fine-tuning without these contrastive samples. Furthermore, when the penalty weight $\lambda$ is increased to more strongly discourage harmful responses, the fine-tuned Llama7b model (under both loss functions) exhibited 'non-English answer' phenomena, which were not observed in the previous experiments. This observation suggests that fine-tuning with LLM-generated seemingly toxic prompts and responses can degrade the model's language performance and is consistent with observations about the use of AI generated data in recent works (Shumailov et al., 2023).

## 6    CONCLUSION AND LIMITATIONS

Our method enables safe LLM responses to toxic prompts during the SFT stage, improving upon prior work by using far less safety-relevant data and only requiring readily available unsafe responses. A key novelty is our use of EMD loss with a cosine distance metric and a novel lower bound for tractable optimization. Despite these improvements, over-alignment persists—consistent with past work—and our findings also highlight risks in training with AI-generated data. We acknowledge that our results are limited by the LLM sizes we can handle, and hope larger-scale evaluations can be conducted by industry or large consortiums.

ETHICAL STATEMENT

There are dangers and limitations with our study. While we have taken extensive precautions, there is a possibility that some of the prompts and outputs we produce and release could be misused or lead to unsafe outcomes. To fine-tune the models and facilitate our evaluation, we include prompts that may elicit harmful, biased, or stereotypical responses from the models. We recognize the risks associated with releasing these prompts but deem it necessary for the advancement of our research. Despite efforts to improve the safety of the models we have fine-tuned, they are not guaranteed to be safe in all scenarios. Certain edge cases may still result in inappropriate or harmful content generation. Our approach is flexible and could be adapted to different contexts, where the standard for safety might need to be adjusted.

ACKNOWLEDGMENT

This research/project is supported by the National Research Foundation Singapore and DSO National Laboratories under the AI Singapore Programme (AISG Award No: AISG2-RP-2020-017).

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
