# A    APPENDIX

## A.1    FINE-TUNING DETAILS

We follow Safety-Tuned-LLamas (STL) (Bianchi et al., 2023) to use the same prompt template to train all the models described in the paper (Llama 7b, Llama 13b, and Mistral 7b and Llama3.1 8b):

*Below is an instruction that describes a task, paired with an input that provides further context. Write a response that appropriately completes the request.*

### Instruction: {instruction}

### Input: {input}

### Response:

The base models we use are available on HuggingFace. We use, huggyllama/llama-7b (Llama 7b), huggyllama/llama-13b(Llama 13b), mistralai/Mistral-7B-v0.3(Mistral7b) and meta-llama/Meta-Llama-3.1-8B(Llama3.1 8b).

## A.2    HYPER PARAMETERS

All models have been trained NVIDIA L40 or H100 GPUs. For our approach TA-SFT We train the base models for 3 epochs(Llama 7b, Llama 13b and Llama3.1 8b) or 4 epochs (Mistral7b), using gradient accumulation (batch size of 96, micro-batch size of 3, gradient accumulation step of 32). The learning rate is set to 1e-4 for all models. The parameters for low-rank adaptations are as follows. Alpha is 16, dropout is set to 0.05 and r is set to 8. Target modules are [q_proj,v_proj]. We use grid search to tune the penalty weight $\lambda$. The tuned EMD and NLCL penalty weights for LLMs fine-tuned with 1,000, 500, 300, and 100 toxic prompts are shown in the Table 4.

Table 4: The penalty weight $\lambda$ for our TA-SFT approach with EMD and NLCL loss.

|  | # Toxic | Llama7b | Llama13b | Mistral7b | Llama3.1-8b |
|---|---|---|---|---|---|
| **EMD** | 1000 | 0.83 | 0.70 | 0.50 | 0.49 |
|  | 500 | 1.70 | 0.99 | 0.60 | 0.78 |
|  | 300 | 4.00 | 2.20 | 1.05 | 1.30 |
|  | 100 | 9.00 | 7.10 | 3.10 | 3.80 |
| **NLCL** | 1000 | 3.80 | 5.50 | 2.40 | 2.50 |
|  | 500 | 4.00 | 12.00 | 3.40 | 3.50 |
|  | 300 | 14.50 | 15.00 | 5.60 | 5.50 |
|  | 100 | 20.00 | 55.00 | 25.00 | 16.00 |

## A.3    PROOF OF PROPOSITION 1

*Proof.* Note that a simple application of Cauchy Schwarz inequality $n$ times yields the result that $n \sum_{i=1}^{n} ||x_i||^2 \geq || \sum_{i=1}^{n} x_i||^2$ for $n$ vectors $x_i$. We use this fact below. Let $\gamma$ be the joint distribu-

tion (coupling) that is the minimizer in the definition of EMD.

$$\text{EMD}(P, Q_\theta; d_c) = \sum_{x \in V} \sum_{y \in V} \gamma(x, y) d_c(\hat{e}_x, \hat{e}_y)$$

$$= \frac{1}{2} \sum_{x \in V} \sum_{y \in V} \gamma(x, y) \|\hat{e}_x - \hat{e}_y\|^2$$

$$\geq \frac{1}{2} \sum_{x \in V} \sum_{y \in V} (\gamma(x, y))^2 \|\hat{e}_x - \hat{e}_y\|^2 \qquad \text{as } \gamma(x, y) \leq 1, \text{ so } \gamma(x, y) \geq (\gamma(x, y))^2$$

$$= \frac{1}{2} \sum_{x \in V} \sum_{y \in V} \|\gamma(x, y)\hat{e}_x - \gamma(x, y)\hat{e}_y\|^2$$

$$\geq \frac{1}{2|V|^2} \|\sum_{x \in V} \sum_{y \in V} \gamma(x, y)\hat{e}_x - \sum_{x \in V} \sum_{y \in V} \gamma(x, y)\hat{e}_y\|^2 \qquad \text{as } n \sum_{i=1}^{n} \|x_i\|^2 \geq \|\sum_{i=1}^{n} x_i\|^2$$

$$= \frac{1}{2|V|^2} \|\sum_{x \in V} P(x)\hat{e}_x - \sum_{y \in V} Q_\theta(y)\hat{e}_y\|^2 \qquad \text{as } P, Q \text{ are marginals of } \gamma$$

$\square$

## A.4 ADDITIONAL RESULTS

### A.4.1 COMPLEXITY OF EMD COMPUTATION

The additional training time required for our TA-SFT method is minimal, amounting to only 1–2% longer than that of Original SFT. This ensures that TA-SFT remains scalable to very large datasets. Below, we detail the modest computational requirements of TA-SFT:

TA-SFT introduces an additional loss term based on the Earth Mover's Distance (EMD). Computing the EMD term involves two matrix multiplications and one squared Euclidean distance calculation, all of which are efficiently executed on GPUs. Once the EMD term is computed, the backpropagation process in TA-SFT is identical to that of Original SFT, and the forward pass remains unchanged. Consequently, the computational overhead introduced by TA-SFT is negligible.

We conducted experiments to measure the training time for both SFT and TA-SFT using consistent hardware and configurations:

- LLaMA-7B: Trained on NVIDIA L40 GPUs.
- LLaMA-13B: Trained on NVIDIA H100 96GB GPUs.

All experiments used the same batch size, gradient accumulation steps, and training for 3 epochs. The key difference is that SFT was trained on 20k Alpaca instruction-following data, while TA-SFT included an additional 1k unsafe (toxic prompt, harmful response) pairs, leading to slightly longer total training steps for TA-SFT. As shown in the Table 5, the Average Training Time per Step indicates that TA-SFT is only 1.17% slower for LLaMA-7B and 2.36% slower for LLaMA-13B compared to SFT.

Table 5: The comparison of average training time per step between TA-SFT with EMD term and standard SFT.

| Model | Supervised Finetuning (SFT) | | TA-SFT | |
|---|---|---|---|---|
| | Total Training Time (s) | Average Training Time per Step (s) | Total Training Time (s) | Average Training Time per Step (s) |
| Llama-7B | 3342±2 | 5.3590 | 3346±4 | 5.4225 |
| Llama-13B | 2729±10 | 4.3696 | 2927±7 | 4.4729 |

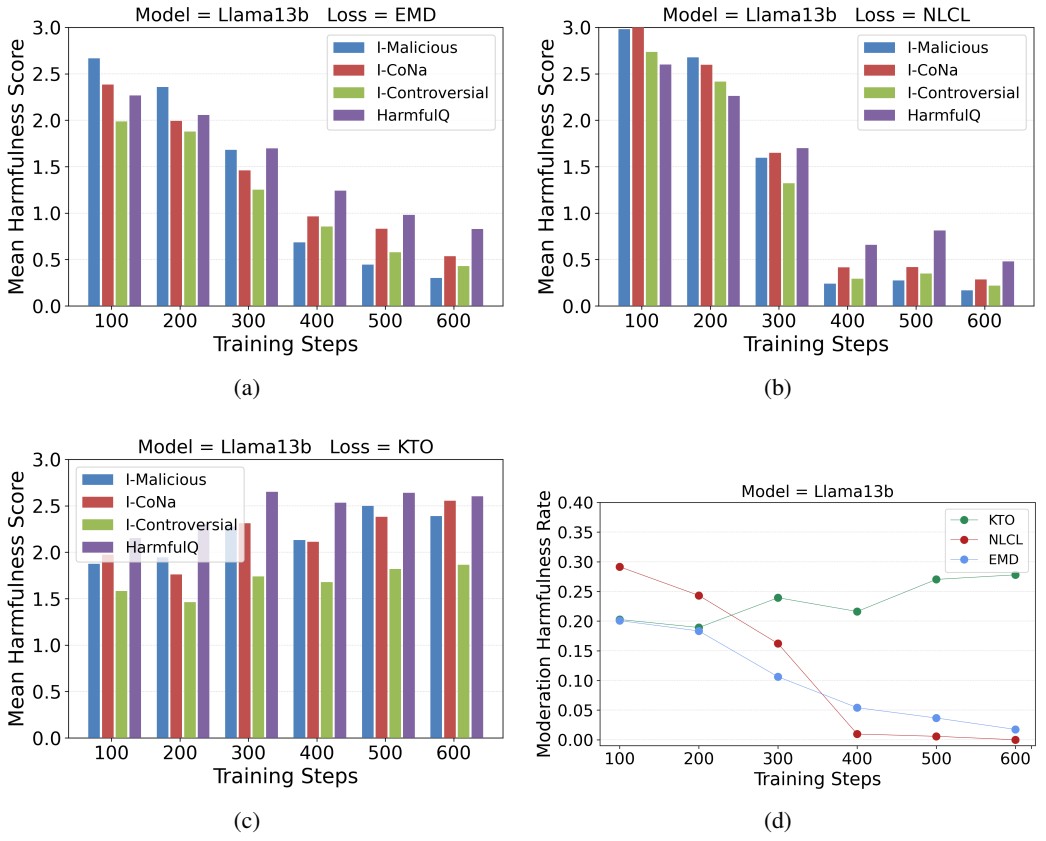

Figure 6: Response safety evaluation on four harmfulness benchmarks for Llama 13b. (a)(b)(c) The mean DeBERTa harmfulness score for KTO and our TA-SFT approach with EMD loss and NLCL loss, separately. Lower scores indicate less harmful (safer) responses. (d) The OpenAI Moderation harmful rate.

### A.4.2 SAFETY LEVEL OF LLAMA 13B, MISTRAL 7B AND LLAMA3.1 8B

To confirm our results, we also tested our TA-SFT with EMD loss and NLCL loss on Llama 13b (Figure 6), Mistral 7b (Figure 7) and Llama3.1 8b (Figure 8). These figures present both the harmfulness score from DeBERTa model and the harmfulness percentage from OpenAI moderation API. All models exhibit similar to those observed for the Llama 7b model in Section 5.3.1 of the main paper, showing a decrease in harmfulness as training progresses using our TA-SFT method, while KTO fails to improve safety levels. Moreover, our TA-SFT approach, with both EMD loss and ORPO loss, ultimately reduces the harmfulness rate to nearly 0%.

As stated in the main paper, the OpenAI Moderation API also provide a harmful score beside a binary tag which are shown in Figure 9. The curves in Figure 9 representing the average harmfulness score across all responses in the four harmfulness benchmarks, exhibit a similar trend to the harmfulness rates from the OpenAI Moderation API, depicted in Figure 2, Figure 6, Figure 7, Figure 8,

### A.4.3 RESPONSE QUALITY

To substantiate the claim that fine tune LLMs with our TA-SFT using both EMD and NLCL loss does not degrade response quality (Section 5.3.2), we additionally evaluated the response quality on PIQA and OpenBookQA shown in Table 6.

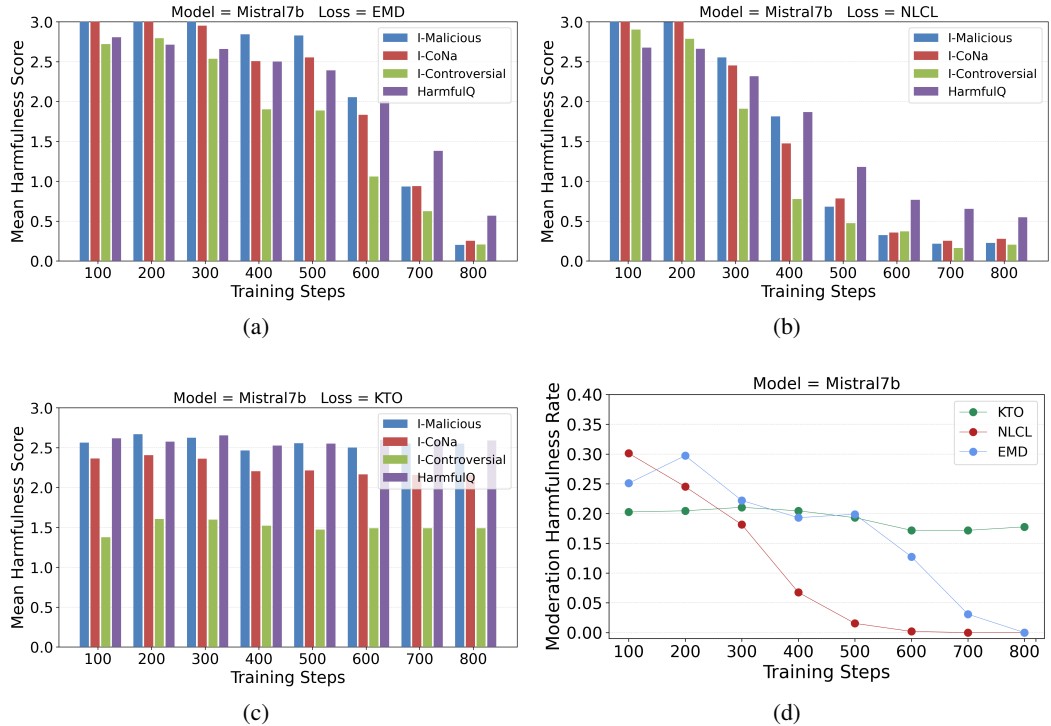

Figure 7: Response safety evaluation on four harmfulness benchmarks for Mistral 7b. (a)(b)(c) The mean DeBERTa harmfulness score for KTO and our TA-SFT approach with EMD loss and NLCL loss, separately. Lower scores indicate less harmful (safer) responses. (d) The OpenAI Moderation harmful rate.

Table 6: The response quality of four tested models on additional two multi-choice language modeling benchmarks. There are not degrading patterns in terms of performance from our TA-SFT approach with EMD loss and NLCL loss.

| Model | PIQA | | | | OpenBookQA | | | |
|---|---|---|---|---|---|---|---|---|
| | SFT | KTO | NLCL | EMD | SFT | KTO | NLCL | EMD |
| **Llama7b** | 77.09 | 89.11 | 79.27 | 79.22 | 32 | 35.4 | 35.2 | 34.8 |
| **Llama13b** | 75.46 | 79.11 | 79.33 | 79.33 | 35.6 | 34.8 | 34 | 33.4 |
| **Mistral7b** | 77.31 | 80.85 | 81.23 | 80.85 | 34 | 35.6 | 33.8 | 33.8 |
| **Llama3.1-8b** | 80.32 | 80.96 | 80.14 | 80.41 | 35 | 37 | 35.2 | 35.2 |

### A.4.4 Data Efficiency: Fewer Harmful Examples

To confirm the statement that we made in Section 5.3.3, we present the number of harmful responses across the four harmfulness benchmarks in Table 7, using our TA-SFT approach with EMD and NLCL. The EMD loss function enables LLMs to learn safe responses with only 100 harmful examples on these two models, whereas the NLCL loss function fails to achieve this.

### A.4.5 Training Data: Safe Samples vs Unsafe Samples

To confirm the observation that we made in Section 5.3.4, we compare the performance of Safety-Tuned Llamas (STL) with our TA-SFT approach using EMD loss on Mistral7b and Llama3.1 8b, despite the latter being fine-tuned with a smaller number of harmful data. Although STL benefits from high-quality safe responses to toxic prompts, it is evident that TA-SFT with EMD loss still significantly outperforms STL(Table 8).

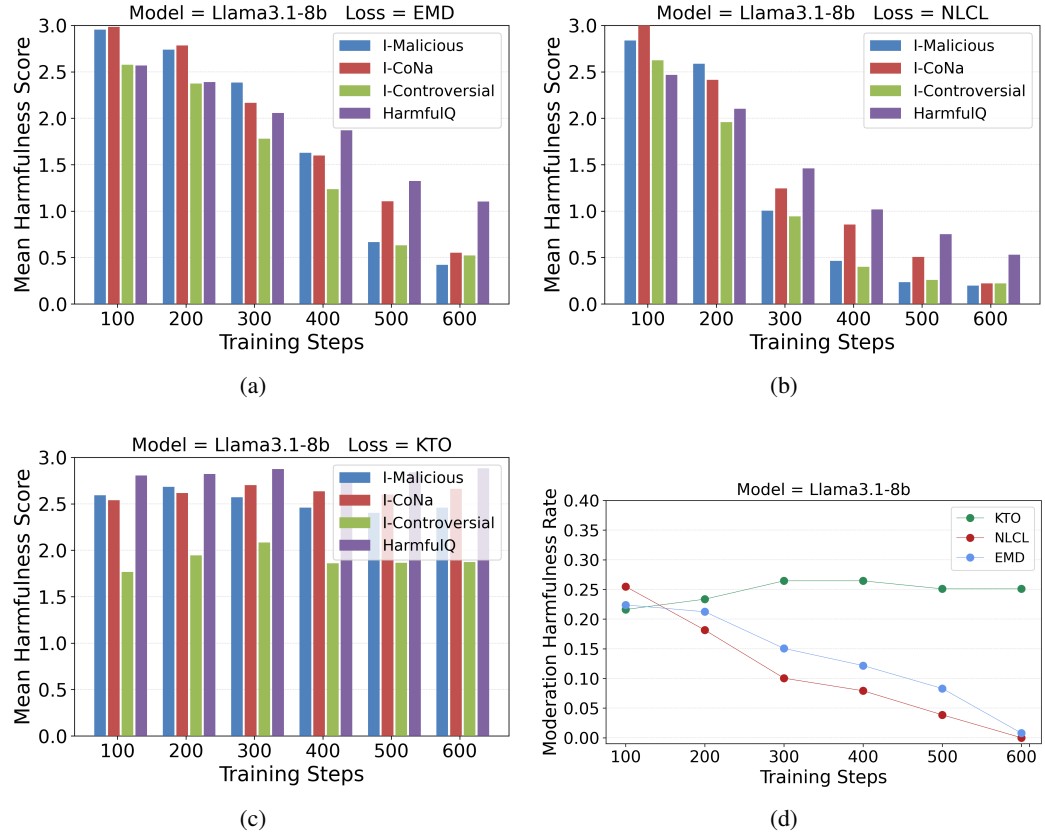

Figure 8: Response safety evaluation on four harmfulness benchmarks for Llama3.1 8b. (a)(b)(c) The mean DeBERTa harmfulness score for KTO and our TA-SFT approach with EMD loss and NLCL loss, separately. Lower scores indicate less harmful (safer) responses. (d) The OpenAI Moderation harmful rate.

Table 7: Number of harmful responses using EMD and NLCL losses with fewer toxic prompts. EMD loss exhibits higher data-efficiency to make LLMs achieve high safety level (lower number of harmful responses) with only 100 toxic prompts in the instruction-tuning dataset.

| Model | # Toxic | I-Malicious | | I-CoNa | | I-Controversial | | HarmfulQ | |
|---|---|---|---|---|---|---|---|---|---|
| | | NLCL | EMD | NLCL | EMD | NLCL | EMD | NLCL | EMD |
| **Mistral 7b** | 1000 | 0 | 0 | 0 | 0 | 0 | 0 | 0 | 0 |
| | 500 | 1 | 1 | 1 | 2 | 0 | 0 | 0 | 0 |
| | 300 | 1 | 1 | 8 | 5 | 0 | 0 | 0 | 1 |
| | 100 | 3 | 0 | 53 | 4 | 2 | 0 | 4 | 1 |
| **Llama3.1-8b** | 1000 | 0 | 0 | 0 | 1 | 0 | 0 | 0 | 3 |
| | 500 | 1 | 0 | 6 | 6 | 2 | 2 | 1 | 1 |
| | 300 | 1 | 0 | 6 | 5 | 0 | 2 | 1 | 1 |
| | 100 | 1 | 0 | 12 | 7 | 2 | 2 | 1 | 1 |

### A.4.6 OVER-ALIGNMENT OF MISTRAL 7B AND LLAMA3.1 8B

Consistent with the observation in Section 5.3.5 for Mistral 7b and Llama3.1 8b, as illustrated in Figure 10, over-refusal issues do not emerge at the beginning of training for Llama 7b and Llama 13b with NLCL and EMD, despite the relatively low safety levels. However, as training progresses,

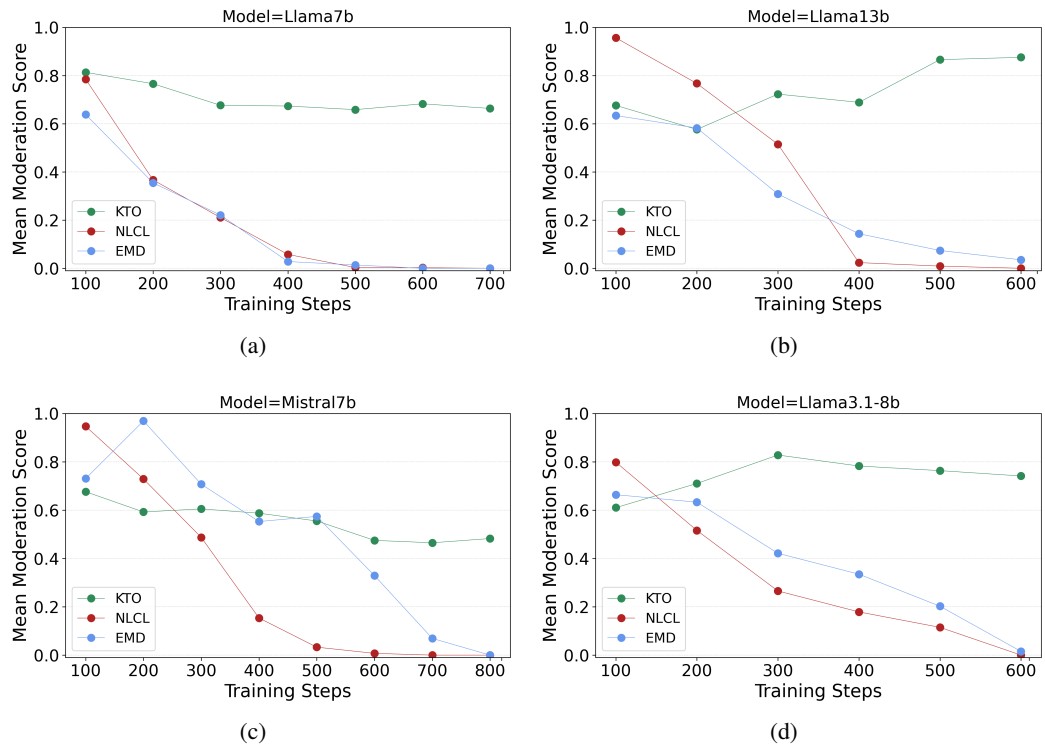

Figure 9: The averaged OpenAI Moderation harmful scores for KTO and our TA-SFT approach with EMD loss and NLCL loss.

Table 8: Number of harmful responses using EMD and safety-tuned-llamas (STL) Bianchi et al. (2023) with fewer toxic prompts. There is a notable increase in the number of harmful responses (indicating a decrease in safety) for STL as the number of safe responses in the instruction-tuning dataset decreases.

| Model | # Toxic | I-Malicious STL | I-Malicious EMD | I-CoNa STL | I-CoNa EMD | I-Controversial STL | I-Controversial EMD | HarmfulQ STL | HarmfulQ EMD |
|---|---|---|---|---|---|---|---|---|---|
| **Mistral 7b** | 1000 | 0 | 0 | 0 | 0 | 0 | 0 | 1 | 0 |
| | 500 | 0 | 1 | 1 | 2 | 0 | 0 | 0 | 0 |
| | 300 | 1 | 1 | 13 | 5 | 0 | 0 | 1 | 1 |
| | 100 | 8 | 0 | 64 | 4 | 1 | 0 | 5 | 1 |
| **Llama3.1-8b** | 1000 | 0 | 0 | 0 | 1 | 0 | 0 | 1 | 3 |
| | 500 | 1 | 0 | 7 | 6 | 0 | 2 | 1 | 1 |
| | 300 | 2 | 0 | 22 | 5 | 0 | 2 | 3 | 1 |
| | 100 | 11 | 0 | 71 | 7 | 1 | 2 | 5 | 1 |

both NLCL and EMD improve the safety of the LLMs but also result in an increased occurrence of over-refusal.

### A.4.7 LARGER FINETUNING DATASET

We conducted further evaluation of our models trained with a larger dataset to explore the impact of increased data size on performance. We expanded the fine-tuning dataset to 2.5 times larger than the dataset used in the main paper, resulting in a total of 50k data samples from Alpaca and 2,500 unsafe (toxic prompts and unsafe responses) pairs.

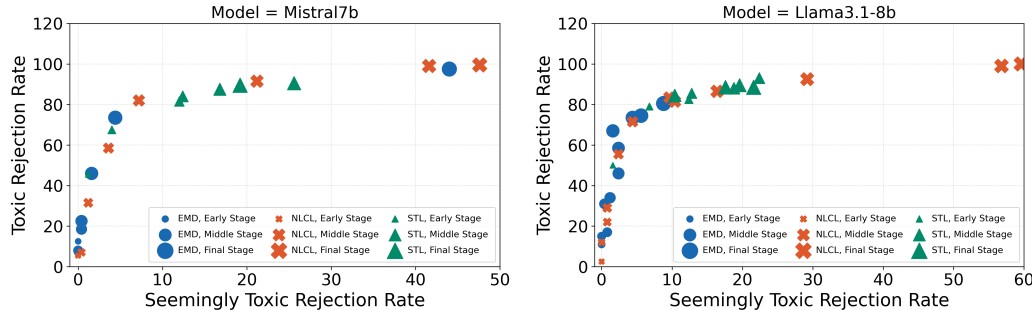

Figure 10: Over-refusal vs. Safety Levels at different training Stages for Mistral 7b and Llama3.1 8b Models. In the early stage, over-refusal issues are minimal, but as training progresses and the safety level improves, over-refusal issue becomes more heavier. Both TA-SFT and STL show the same trend, empirically demonstrating that the inclusion of refusal examples in the instruction-following dataset is not the cause of the over-refusal issue.

Table 9: Performance of TA-SFT with EMD term trained with larger dataset.

| Model | Finetuning data | OR-Bench | AlpacaEval |
|---|---|---|---|
| Llama-7B | 50k+2500 unsafe examples | 0 | 57.22 |
| Llama-7B | 20k+1000 unsafe examples | 1 | 57.37 |
| Llama-13B | 50k+2500 unsafe examples | 2 | 62.35 |
| Llama-13B | 20k+1000 unsafe examples | 5 | 62.24 |

As shown in Table 2 of the main paper, our TA-SFT method with EMD term already achieves peak performance on the four safety evaluation benchmarks (with 0 harmful responses). To further challenge our method, we evaluated its performance on the larger, more diverse and newly released OR-Bench-Toxic benchmark (Cui et al., 2024). This benchmark includes 655 toxic prompts across 10 toxic types, providing broader coverage and a more rigorous evaluation. We use the following metric to evaluate the performance.

- Safety Level: Measured by the number of harmful responses (lower is better).
- Response Quality: Measured using the same method and settings on the AlpacaEval benchmark as in the main paper (higher is better).

As shown in the Table 9, models trained with the larger dataset achieved:

- Slightly better safety levels on the OR-Bench-Toxic benchmark.
- Comparable response quality to models trained on the original dataset.

These results demonstrate that increasing the dataset size can further enhance the model's safety levels without compromising response quality. This finding suggests that scaling the fine-tuning dataset is a promising approach for improving safety in large language models.

### A.4.8 EVALUATION WITH JAILBREAKING ATTACKS

To further evaluate the robustness of the proposed TA-SFT method, We compared the performance of TA-SFT against the baseline approach, Safety-Tuned LLaMAs (STL) under the attacking of jailbreaking. We followed prior work (Chao et al., 2023) to implement the jailbreaking which requires the following three components:

- Attacker LLM: GPT-4O generates jailbreaking prompts.
- Target LLM: Models trained with our method.
- Judging LLM: GPT-4O evaluates responses and scores their harmfulness on a scale of 1 to 10, where a score of 10 indicates a successful attack. Lower scores signify less harmfulness and better robustness against jailbreaking.

Table 10: The comparison of Attack Success Rate (ASR) and Mean Judge Score between our method TA-SFT (Ours) and Safety-tuned-llamas (STL).

| Method | Finetuning Data | ASR | Mean Judge Score |
|--------|-----------------|-----|------------------|
| Ours | Alpaca+1000 unsafe examples | **19.57%** | **6.02** |
| Ours | Alpaca+500 unsafe examples | 26.09% | 7.39 |
| Ours | Alpaca+300 unsafe examples | 34.78% | 8.11 |
| Ours | Alpaca+100 unsafe examples | 63.04% | 9.13 |
| STL | Alpaca+1000 safe examples | 60.00% | 8.91 |
| STL | Alpaca+500 safe examples | 63.04% | 8.71 |
| STL | Alpaca+300 safe examples | 73.33% | 9.33 |
| STL | Alpaca+100 safe examples | 69.57% | 9.07 |

The results, summarized in the Table 10, demonstrate that our method significantly outperforms STL in both Attack Success Rate, ASR (lower is better) and Mean Judge Score (lower is better). Notably, our method achieves a 19.57% ASR and a 6.02 Mean Judge Score, indicating superior robustness. Additionally, the robustness improves as the number of unsafe examples used during training increases.