# OpenReview forum: "Semantic Loss Guided Data Efficient Supervised Fine Tuning for Safe Responses in LLMs"
_ICLR.cc/2025/Conference — ICLR 2025 Poster_

### Official Review · Reviewer_VgDz · 2024-10-23

**Soundness:** 3
**Presentation:** 3
**Contribution:** 3
**Rating:** 6
**Confidence:** 3

**Summary:**

This paper proposes an EMD penalty term to prevent LLM from generating harmful information. The function of this penalty term is to make the probability distribution of tokens generated by LLM in the case of toxic prompts far away from the probability distribution of tokens in harmful responses dataset. When used for safety alignment, this penalty term allows SFT not to rely on a large number of human-collected SFT datasets while achieving a comparable or even better performance. And it will not cause the problem of over-alignment.

**Strengths:**

(1) This paper takes a novel approach by focusing on the perspective of distancing from harmful responses, differing from previous methods that directly align responses.

(2) This paper takes a novel approach by focusing on distancing from harmful responses, which differs from previous methods that directly align responses. Experimental results indicate that this method achieves comparable or even superior performance using a dataset size significantly smaller than that of previous methods, demonstrating the effectiveness of the approach.

**Weaknesses:**

(1) A detailed explanation is required for how the probability distribution represented by P (·|w<t−1) in line 186 of the paper is obtained. After reading the paper, I don't understand what kind of probability distribution it is.

(2)The paper discusses that good performance can be achieved with a small safety-related dataset. So, would increasing the dataset size lead to even higher performance?

**Questions:**

see weaknesses

---

> ### Author Response · Authors · 2024-11-18
>
> We sincerely thank the reviewer for their valuable comments and insights. Below, we address the points raised in the review.
>
> # Explanation Needed for How P(⋅∣w\<t−1) is Derived.
>
> We thank the reviewer for carefully reading our method and raising this important question. To address the concern, we would like to guide the reviewer to the **implementation details** provided in Line 210 of the paper. In this section, we explain that we treat P as a one-hot vector of the next token as present in the safety related dataset. We greatly appreciate the reviewer’s feedback and will revise this section to improve its readability.
>
> # Increasing the Finetuning Dataset Size
>
> We conducted an additional evaluation of our models trained with a larger dataset   to explore the impact of increased data size on performance. Limited by the available instruction-following data in the Alpaca dataset, we expanded the fine-tuning dataset to **2.5 times its original size**, resulting in a total of 50k data samples from Alpaca and 2,500 unsafe (toxic prompts and unsafe responses) pairs.
>
> As shown in **Table 2** of the main paper, our TA-SFT method with EMD loss already achieves peak performance on the four safety evaluation benchmarks (with **0 harmful responses**). To further challenge our method, we evaluated its performance on the larger, more diverse and newly released **OR-Bench-Toxic** benchmark. This benchmark includes **655 toxic prompts** across **10 toxic types**, providing broader coverage and a more rigorous evaluation. We use the following metric to evaluate the performance.
>
> * **Safety Level**: Measured by the number of harmful responses (lower is better).
> * **Response Quality**: Measured using the same method and settings on the **AlpacaEval** benchmark as in the main paper (higher is better).
>
> As shown in the table below, models trained with the larger dataset achieved:
>
> 1. Slightly better safety levels on the OR-Bench-Toxic benchmark.
> 2. Comparable response quality to models trained on the original dataset.
>
> These results demonstrate that increasing the dataset size can further enhance the model’s safety levels without compromising response quality. This finding suggests that scaling the fine-tuning dataset is a promising approach for improving safety in large language models.
>
> | Model | Finetuning data | OR-Bench | AlpacaEval |
> | :---- | :---- | :---- | :---- |
> | Llama-7B | 50k+2500 unsafe examples | 0 | 57.22 |
> | Llama-7B | 20k+1000 unsafe examples | 1 | 57.37 |
> | Llama-13B | 50k+2500 unsafe examples | 2 | 62.35 |
> | Llama-13B | 20k+1000 unsafe examples | 5 | 62.24 |
>
>
>
> *We will update the PDF with our new results and typo fixes after performing all experiments asked by all reviewers.*

---

### Official Review · Reviewer_hhQn · 2024-11-01

**Soundness:** 3
**Presentation:** 3
**Contribution:** 3
**Rating:** 6
**Confidence:** 3

**Summary:**

This paper proposes a method to improve safety in large language models (LLMs) by utilizing a small, easily obtainable set of unsafe responses. Unlike approaches relying on extensive human feedback or corrective data from other LLMs, this method applies an Earth Mover Distance (EMD) based semantic loss to guide the LLM away from generating unsafe responses. Additionally, the paper proposes a novel lower bound for EMD to optimize efficiency. Experiments with multiple LLM models demonstrate improved safety and response quality with reduced data requirements compared to baselines. The method also addresses over-alignment issues that often arise with refusal-based safe responses.

**Strengths:**

- Data Efficiency: The method requires minimal safety-related data, making it cost-effective and accessible. The use of EMD loss may be a reasonable way to conduct alignment based on the token semantics, rather than only on the output.
- Semantic Loss Innovation: The application of EMD-based semantic loss introduces a novel, context-sensitive approach to penalizing unsafe responses.
- Maintained Response Quality: The approach maintains response quality across reasoning and conversational benchmarks.

**Weaknesses:**

- When calculating the EMD loss, the entire distribution of the LLM vocabulary is considered. However, there are also a lot of tokens in the harmful responses that may not be harmful if treated separately. I wonder whether maximizing the distance would result in undesirable outcomes that affect the original capability of LLM. If so, would first extract harmful keywords from the responses and only calculate the EMD loss using those tokens help?
- If the number of safety related QA pairs is limited, could the performance of SFT be improved by simply upscaling the loss for safety related QA pairs, or through oversampling?
I will consider raising the score if the authors help me with those concerns.

**Questions:**

See weaknesses.

---

> ### Author Response · Authors · 2024-11-18
>
> We sincerely thank the reviewer for their valuable comments and insights. Below, we address the points raised in the review.
>
> # Calculating EMD on Part of the Unsafe Responses
>
> The idea of focusing on specific tokens in harmful responses when calculating the EMD loss is indeed attractive. However, defining what constitutes a "harmful word" is quite challenging. For instance, while the word "kill" may generally be considered harmful, in a context such as "how to kill a python process," it is not harmful. This context-dependence complicates the process of reliably identifying harmful keywords.
>
> Yet, taking inspiration from the reviewer’s suggestion, we consider penalizing only parts of a response. To address this idea in a simpler and more targeted way, we focused on the beginning of the response. Intuitively, the initial words of an LLM's response often set the tone or "attitude" of the entire reply, determining whether it refuses to answer or engages with the prompt. In this additional experiment, we only penalize the first 5 words of the response, aiming to influence the LLM's overall attitude toward toxic prompts while minimizing unintended effects on its original capabilities.
>
> | Model | The Whole Sentence |  | The First 5 Words |  |
> | :---- | :---- | :---- | :---- | :---- |
> |  | I-CoNa | AlpacaEval | I-CoNa | AlpacaEval |
> | Llama-7B | 0 | 57.37 | 1 | 50.3 |
> | Llama-13B | 0 | 62.24 | 0 | 57.93 |
>
> We evaluated the safety levels of LLaMA-7B and LLaMA-13B using the same fine-tuning dataset described in the main paper, measured on the I-CoNa benchmark, and recorded the corresponding response quality. Our findings indicate that achieving comparable safety levels in the fine-tuned models leads to a significant decrease in response quality if only penalize the first 5 words. During the experiments, we observed that when penalizing only the first 5 words of the response, the weight of the EMD term must be adjusted to a much higher value to reach a comparable safety level. Intuitively, this is because penalizing only the first 5 words involves a smaller subset of tokens compared to penalizing the entire response. But a high penalty weight would cause the next token prediction distribution of some tokens to change significantly. We believe that increasing the diversity and quantity of toxic prompts and their corresponding harmful responses in the fine-tuning dataset could mitigate this issue. This represents an interesting direction for future exploration.
>
> # Upscaling the Loss for Safety Related QA Pairs
>
> We would like to clarify that in the standard SFT approach, the model is trained only on **safety-unrelated datasets**, which do not include toxic prompts or harmful responses. In contrast, our proposed TA-SFT method is trained on a **mixed dataset** comprising both safety-unrelated datasets and **safety-related datasets**. To that end, in our TA-SFT method, besides the SFT term in the loss, there is an additional EMD term (or NLCL term) to penalize the harmful responses. A weight $\\lambda$ is multiplied with the EMD or NLCL term. Adjusting this weight $\\lambda$ is equivalent to upscaling the loss for safety related QA pairs. We have tuned these weights for both the EMD term and NLCL term to achieve extremely high safety levels, as demonstrated by the results on four benchmarks in **Table 2** and **Figure 2** in the pape**r**.
>
> *We will update the PDF with our new results and typo fixes after performing all experiments asked by all reviewers.*

---

> > ### Author Response · Authors · 2024-11-25
> >
> > We realize that we answered the reviewer's question earlier from the perspective of our approach, and would like to provide a further response to the query about "Upscaling the Loss for Safety-Related QA Pairs” from the perspective of the prior work Safety-Tuned LLaMAs (STL).
> >
> > For Safety-Tuned LLaMAs (STL), the model is trained using standard supervised fine-tuning (SFT) loss on a mix of safety-unrelated data and safety-related (toxic prompt, safe response) pairs. As shown in Table 3 of the main paper, STL's performance declines significantly as the number of safety-related pairs decreases from 1,000 to 500, 300, and 100\. In the context of STL, we interpret the reviewer’s question as: could STL performance be improved by oversampling safety-related pairs or by upscaling the loss specifically for these pairs during training?
> >
> > To explore this, we conducted experiments using LLaMa-7B trained with 500 and 100 safety-related (toxic prompt, safe response) pairs, applying oversampling rates ranging from 1x to 8x. The number of harmful responses to each safety benchmark are presented in the table below. In the "Method" column of the table, "STL-x" indicates that LLaMA-7B was trained with an oversampling rate of x, and ‘Ours’ indicate our proposed TA-SFT method. Additionally, we evaluated the models using a recently released safety benchmark, OR-Bench-Toxic, which contains 655 toxic prompts across 10 distinct categories of toxic input.
> >
> > Our findings indicate that oversampling safety-related pairs does provide some improvement to STL performance. However, our proposed TA-SFT approach continues to significantly outperform STL, even with a large oversampling rate.
> >
> > | Method | \# Safe-Related Data | I-MaliciousInstructions | I-CoNa | I-Controversial | HarmfulQ | OR-Bench |
> > | :---- | :---- | :---- | :---- | :---- | :---- | :---- |
> > | STL-1 | 100 | 6 | 60 | 3 | 5 | 126 |
> > | STL-2 | 100 | 4 | 44 | 5 | 4 | 88 |
> > | STL-3 | 100 | 5 | 22 | 3 | 4 | 83 |
> > | STL-4 | 100 | 1 | 19 | 0 | 5 | 80 |
> > | STL-5 | 100 | 5 | 13 | 1 | 3 | 76 |
> > | STL-6 | 100 | 2 | 9 | 3 | 3 | 86 |
> > | STL-7 | 100 | 2 | 9 | 1 | 3 | 79 |
> > | STL-8 | 100 | 2 | 8 | 0 | 2 | 75 |
> > | **Ours** | 100 | **0** | **5** | **0** | **0** | **23** |
> > | STL-1 | 500 | 1 | 16 | 3 | 3 | 76 |
> > | STL-2 | 500 | 3 | 8 | 1 | 3 | 69 |
> > | STL-3 | 500 | 3 | 7 | 1 | 1 | 61 |
> > | STL-4 | 500 | 3 | 7 | 1 | 2 | 55 |
> > | STL-5 | 500 | 2 | 6 | 0 | 3 | 52 |
> > | STL-6 | 500 | 2 | 5 | 0 | 2 | 57 |
> > | STL-7 | 500 | 1 | 6 | 0 | 3 | 61 |
> > | STL-8 | 500 | 2 | 5 | 0 | 4 | 51 |
> > | **Ours** | 500 | **0** | **0** | **0** | **1** | **2** |
> > |  |  |  |  |  |  |  |

---

> > > ### Comment · Reviewer_hhQn · 2024-11-26
> > > **Response to authors**
> > >
> > > I appreciate the efforts made by the authors during their rebuttal.
> > > - I am OK with the first response, since it is indeed difficult to tell which tokens are harmful in many cases.
> > > - With the second response, I think the oversampling counterpart does not seem to saturate yet, it seems to be improving. I wonder what is the number of safety-unrelated instructions in the training data? I think the reason why this approach is not performing well might be that the impact of safety-related instructions is diminished due to the scale difference. I think making the sampling rate higher such that safety-unrelated and safety-related instructions are comparable would potentially solve this issue.
> > >
> > > Even though my concerns are not yet fully addressed, I tend to increase my score, since this paper proposes an interesting strategy to mitigate safety concerns of LLMs.

---

> > > > ### Author Response · Authors · 2024-11-28
> > > >
> > > > Thank you for your thoughtful feedback and for considering raising your score.
> > > >
> > > > To address your insightful concerns, we conducted additional experiments as follows:
> > > >
> > > > 1. **Higher Sampling Rate for Safety-Related Data:**
> > > >    There are totally 20k safety-unrelated data in the finetuning dataset. We adjusted the sampling rate to 40 times higher for safety-related instructions. This adjustment ensures that the training exposure to safety-related and safety-unrelated instructions is approximately comparable, giving safety-related data a stronger influence during fine-tuning.
> > > > 2. **Exclusive Training on Safety-Related Data:**
> > > >    We also trained LLaMA-7B with only 10k safe responses, completely excluding the 20k safety-unrelated instructions using Safety-tuned-Llamas (STL).
> > > >
> > > > Our results, summarized in the updated table below, indicate that the performance of STL with 100/500 safe-related data, has already reached its peakk, and still **failed to achieve a safety level comparable to our method**. Moreover, when only trained on 100% safety-related dataset, STL still achieve comparable safety level.
> > > > We hope these additional experiments address the concerns. If you have further feedback or suggestions, we would be delighted to discuss them to further improve our work. Thank you again for your valuable input and encouragement.
> > > >
> > > > | Method | \# Safe-Related Data | I-MaliciousInstructions | I-CoNa | I-Controversial | HarmfulQ | OR-Bench |
> > > > | :---- | :---- | :---- | :---- | :---- | :---- | :---- |
> > > > | STL-1 | 100 | 6 | 60 | 3 | 5 | 126 |
> > > > | STL-5 | 100 | 5 | 13 | 1 | 3 | 76 |
> > > > | STL-8 | 100 | 2 | 8 | 0 | 2 | 75 |
> > > > | STL-20 | 100 | 4 | 6 | 0 | 1 | 87 |
> > > > | STL-30 | 100 | 3 | 3 | 0 | 3 | 91 |
> > > > | STL-40 | 100 | 4 | 5 | 0 | 3 | 86 |
> > > > | **Ours** | 100 | **0** | **5** | **0** | **0** | **23** |
> > > > |  |  |  |  |  |  |  |
> > > > |  |  |  |  |  |  |  |
> > > > | STL-1 | 500 | 1 | 16 | 3 | 3 | 76 |
> > > > | STL-5 | 500 | 2 | 6 | 0 | 3 | 52 |
> > > > | STL-8 | 500 | 2 | 5 | 0 | 4 | 51 |
> > > > | STL-20 | 500 | 3 | 2 | 0 | 1 | 64 |
> > > > | STL-30 | 500 | 3 | 4 | 0 | 0 | 58 |
> > > > | STL-40 | 500 | 2 | 5 | 0 | 0 | 57 |
> > > > | **Ours** | 500 | **0** | **0** | **0** | **1** | **2** |
> > > > |  |  |  |  |  |  |  |
> > > > |STL  | 10000 | 2 | 5 | 0 | 1 | 31 |

---

### Official Review · Reviewer_UMZC · 2024-11-01

**Soundness:** 3
**Presentation:** 2
**Contribution:** 3
**Rating:** 6
**Confidence:** 3

**Summary:**

This paper addresses a critical issue in the use of Large Language Models (LLMs)—the generation of unsafe responses to toxic prompts. Traditional methods for mitigating this problem often involve extensive human annotation or the generation of corrective data from other LLMs, which are costly and less reliable. The authors propose a novel method called Toxicity Avoiding Supervised Fine Tuning (TA-SFT), which minimizes the need for large safety-related datasets by utilizing a small set of harmful responses as training data.

**Strengths:**

1.Data Efficiency: The method achieves high safety levels using only 0.5% of the safety-related data required by traditional methods, making it a cost-effective solution.

2. Innovative Use of EMD Loss: The use of EMD as a semantic penalty term in loss functions is novel and effectively discourages the generation of unsafe responses, as it considers semantic similarity.

3. Broad Evaluation: The paper includes extensive evaluation across several LLM architectures (e.g., Llama 7b, Llama 13b, Mistral 7b) and different safety and response quality metrics, demonstrating robustness and generalizability.

**Weaknesses:**

Formatting and Expression Issues:

There are several formatting and expression problems throughout the manuscript. For instance, in line 129, the first occurrence of "Negative Log-Likelihood" requires a citation. In line 120, the introduction of two subscripts needs clarification, even though I can understand it. Additionally, there are multiple instances where punctuation is missing at the end of equations. Lines 248-253 contain incorrect citation formats. Furthermore, some symbols are used without clear definitions. Overall, the manuscript has numerous formatting and expression issues that would benefit from a thorough review by the authors.


Complexity of EMD Computation:

Although EMD improves model safety, it can be computationally intensive. The paper introduces a lower bound to make it tractable, but this might limit scalability for very large datasets.


Limited Over-Alignment Solution:

While the paper identifies over-alignment, it does not fully resolve the issue. Models still show increased refusal rates for benign prompts as training progresses, suggesting room for improvement in managing over-alignment.


Dependence on Specific Safety Benchmark:

The model’s performance is largely benchmarked on safety-related datasets specific to toxic prompts. Its effectiveness in real-world applications or against evolving types of toxic input remains untested.




 Contrastive Data Effect:

The inclusion of AI-generated contrastive samples caused unexpected degradation in response quality, raising concerns about the method's resilience to noise and synthetic data.

Finally, I acknowledge the innovative contributions of this work. Should the authors adequately address the identified issues, I would be inclined to reevaluate my assessment positively.

**Questions:**

See Weaknesses.

---

> ### Author Response · Authors · 2024-11-18
>
> We sincerely thank the reviewer for their valuable comments and insights. Below, we address the points raised in the review.
>
> # Formatting and Expression Issues
>
> We sincerely thank the reviewer for their valuable comments and insights. Below, we address the points raised in the review.
>
> We have conducted a comprehensive review of the manuscript to address all formatting and expression issues, ensuring a polished and professional presentation. We will update the  manuscript once we complete all experiments asked by all reviewers.
>
> # Complexity of EMD Computation
>
> The additional training time required for our TA-SFT method is minimal, only 1–2% longer than that of Original SFT. Therefore, TA-SFT can scale up to a very large dataset. The computational requirements of TA-SFT are modest, as detailed below:
>
> In addition to the original SFT loss, TA-SFT involves calculating the Earth Mover’s Distance (EMD), which requires two matrix multiplications and one squared Euclidean distance calculation. These operations are efficiently performed on GPUs. Once the EMD calculation is complete, the backpropagation in TA-SFT is identical to that of Original SFT. Furthermore, the forward pass in TA-SFT remains the same as in Original SFT. Thus, the additional computational workload introduced by TA-SFT is negligible.
>
> We conducted experiments to measure the training time for both SFT and TA-SFT using consistent hardware and configurations:
>
> * **LLaMA-7B**: Trained on NVIDIA L40 GPUs.
> * **LLaMA-13B**: Trained on NVIDIA H100 96GB GPUs.
>
> All experiments used the same batch size, gradient accumulation steps, and training for 3 epochs. The key difference is that SFT was trained on 20k Alpaca instruction-following data, while TA-SFT included an additional 1k unsafe (toxic prompt, harmful response) pairs, leading to slightly longer total training steps for TA-SFT. As shown in the table below, the **Average Training Time per Step** indicates that TA-SFT is only **1.17% slower for LLaMA-7B** and **2.36% slower for LLaMA-13B** compared to SFT.
>
> | Model | Supervised Finetuning (SFT) |  | TA-SFT |  |
> | :---- | :---- | :---- | :---- | :---- |
> |  | Total Training Time (s) | Average Training Time per Step (s) | Total Training Time (s) | Average Training Time per Step (s) |
> | Llama-7B | 3342±2 | 5.3590 | 3346±4 | 5.4225 |
> | Llama-13B | 2729±10 | 4.3696 | 2927±7 | 4.4729 |
>
> # Limited Over-Alignment Solution
>
> We wish to point out that our work's goal is not to solve the overalignment issue. Instead, our goal is to show that safety can be improved in the SFT stage with a very limited number of safety-related data points, which we demonstrate successfully. Indeed, overalignment remains an open issue that researchers are still struggling to solve.

---

> > ### Author Response · Authors · 2024-11-18
> >
> > # Dependence on Specific Safety Benchmark
> >
> > We appreciate the reviewer’s concern about the applicability of our method to real-world scenarios and evolving types of toxic input.
> >
> > ### Broader Evaluation on Toxic Prompts
> >
> > Evaluating our method on a larger safety benchmark with broader coverage and more diverse types of toxic prompts is indeed valuable. However, due to the limited availability of open-source safety benchmarks, expanding the benchmarks themselves is beyond the scope of this work. Despite these constraints, we have made every effort to ensure that our evaluation is **objective**, **comprehensive**, and **fair** within the available resources.
> >
> > To address this concern of the reviewer further, we conducted additional evaluations using the newly released **OR-Bench-Toxic** dataset, which contains 655 toxic prompts across 10 distinct types of toxic input. We tested the performance of our TA-SFT with EMD loss on this benchmark using both **LLaMA-7B** and **LLaMA-13B**, providing additional insights into the robustness of our approach.
> >
> > | Model | Finetuning data | OR-Bench | I-CoNa |
> > | :---- | :---- | :---- | :---- |
> > | Llama-7B | 20k+1000 unsafe examples | 1 | 0 |
> > | Llama-7B | 20k+500 unsafe examples | 2 | 0 |
> > | Llama-7B | 20k+300 unsafe examples | 0 | 0 |
> > | Llama-7B | 20k+100 unsafe examples | 23 | 5 |
> > | Llama-13B | 20k+1000 unsafe examples | 5 | 0 |
> > | Llama-13B | 20k+500 unsafe examples | 9 | 0 |
> > | Llama-13B | 20k+300 unsafe examples | 7 | 0 |
> > | Llama-13B | 20k+100 unsafe examples | 22 | 1 |
> >
> > ### Real-World Applicability
> >
> > We appreciate the reviewer’s question regarding the real-world applicability of our approach. In an evolving world, we expect to discover a wider variety of toxic prompts and their corresponding harmful responses. This additional data would allow us to iteratively improve the model’s performance by retraining and fine-tuning with our proposed method, further enhancing its robustness and safety over time.
> >
> > However, due to current limitations in benchmark toxic data, we were unable to pursue this iterative improvement process within the scope of this work. Despite this, we believe that our method provides a strong foundation for real-world application, as it minimizes the reliance on large safety-related datasets and costly human-labeled data. Iterative data collection and refinement is an important consideration for real world application, and we are optimistic about our model’s effectiveness in practical settings.
> >
> > # Contrastive Data Effect
> >
> > We would like to emphasize that the primary goal of this paper is to achieve safe alignment during the SFT stage while minimizing the reliance on large safety-related datasets and avoiding the need for costly human-labeled data. The observed degradation in response quality occurs when training LLMs with synthetic safe and unsafe responses to the same prompts. This is just an observation that we report, which does not relate to our primary goal, but is of broader interest. The impact of synthetic data on model performance is a raging issue in LLM research. While some studies suggest that synthetic data can enhance model capabilities, others have demonstrated that it can lead to performance degradation, not only in large language models (LLMs) but also in models like VAE and GMM, sometimes even causing model collapse \[1\]\[2\]\[3\].
> >
> > \[1\] Shumailov I, Shumaylov Z, Zhao Y, et al. AI models collapse when trained on recursively generated data\[J\]. Nature, 2024, 631(8022): 755-759.
> >
> > \[2\] Long L, Wang R, Xiao R, et al. On llms-driven synthetic data generation, curation, and evaluation: A survey\[J\]. arXiv preprint arXiv:2406.15126, 2024\.
> >
> > \[3\] Shumailov I, Shumaylov Z, Zhao Y, et al. The curse of recursion: Training on generated data makes models forget\[J\]. arXiv preprint arXiv:2305.17493, 2023\.
> >
> > *We would be happy to perform any other experiments that the reviewer wishes to see. We will update the PDF with our new results and typo fixes after performing all experiments asked by all reviewers.*

---

> > > ### Comment · Reviewer_UMZC · 2024-11-26
> > >
> > > The experimental section has resolved my concerns effectively.
> > >
> > > Regarding the formatting of the article, the typos I mentioned are only a small portion of the issues. There are still quite a few others, such as the punctuation in Equation 1, which is incorrect. I strongly recommend that you take some time to carefully proofread the entire manuscript. This will greatly enhance the professionalism of your work and make it more appealing to readers.
> > >
> > > That said, I’ve still increased my score to 6. Good luck!

---

> > > > ### Author Response · Authors · 2024-11-28
> > > >
> > > > Thank you for your feedback and for increasing your score. We sincerely appreciate your acknowledgment of the improvements made to address your concerns. We also thank you for pointing out the formatting and typographical issues. We will take your advice seriously and carefully proofread the entire manuscript to ensure it is polished and professional. Your constructive comments have been invaluable, and we are grateful for your engagement and support.

---

### Official Review · Reviewer_L943 · 2024-11-04

**Soundness:** 4
**Presentation:** 4
**Contribution:** 3
**Rating:** 8
**Confidence:** 3

**Summary:**

The paper provides a way for LLMs to respond safely to toxic prompt in the SFT stage without requiring RLHF
and also improves upon prior results by using much less safety relevant data and only require easily available unsafe responses to toxic prompts. A novel EMD based loss is introduced along with an optimizable lower bound.

**Strengths:**

- Very less safety-related samples are required to get safe LLMs
- safe answers are not needed which reduces human effort.
- standard evals are unaffected after training.

**Weaknesses:**

Multiple works question the generalization of current safety training by jailbreaking them [1, 2, 3]. In such a scenario, reducing the number of samples for safety without showing jailbreaking attempts puts a question on the practical applicability of this work.
It would be great to understand how the proposed method generalizes and performs under jailbreaking. I would encourage the authors to explore [3] and try out simple jailbreaking attempts, if needed.

A proper study using safe examples, safe examples + unsafe examples (EMD), and unsafe examples (EMD) (with various sample sizes) to train the model and then using various jailbreak attacks on the LLM to showcase the robustness of the training is required. The overalignment issue pointed out in the paper is a good start.
Results of this experiment:
- Empirically proves the lack of need of safe examples that require costly human annotations.
- We already know that training on safe examples only does not generalize that effectively, this can provide further conclusions.
- Experiments with various sample sizes can answer questions like: lesser #unsafe examples vs more safe examples.

[1] Does Refusal Training in LLMs Generalize to the Past Tense? CoRR abs/2407.11969 (2024)
[2] CodeAttack: Revealing Safety Generalization Challenges of Large Language Models via Code Completion. ACL (Findings) 2024: 11437-11452
[3] https://jailbreakbench.github.io/

**Questions:**

<See weaknesses>

---

> ### Author Response · Authors · 2024-11-18
>
> We sincerely thank the reviewer for their valuable comments and insights. Below, we address the points raised in the review.
>
> # Evaluation with Jailbreaking Attacks
>
> First, we would like to clarify our approach focuses on incorporating safety alignment during the supervised fine-tuning (SFT) stage of large language models (LLMs). This is distinct from traditional defense methods which aim to make LLMs safer after SFT or RLHF. Importantly, the LLMs trained using our method can be further combined with existing defense mechanisms to enhance robustness against jailbreaking attacks.
>
> We appreciate the reviewer's suggestion to use jailbreaking to evaluate the generalization and robustness of our method. Following this recommendation, we selected the jailbreaking method outlined in reference \[3\] provided by the reviewer: *"Jailbreaking Black Box Large Language Models in Twenty Queries"*
>
> We compared the performance of our method against the baseline approach, Safety-Tuned LLaMAs (STL), across various data sizes of unsafe examples (these are not jailbreaking examples). Key differences between the two methods include:
>
> * **TA-SFT with EMD (ours)**: Trained using instruction-following data mixed with unsafe examples.
> * **STL**: Trained using instruction-following data mixed with safe examples.
>
> Jailbreaking method from \[3\] requires the following three components:
>
> * **Attacker LLM**: GPT-4O generates jailbreaking prompts based on  \[3\]
> * **Target LLM**: Models trained with our method.
> * **Judging LLM**: GPT-4O evaluates responses and scores their harmfulness on a scale of 1 to 10, where a score of 10 indicates a successful attack. Lower scores signify less harmfulness and better robustness against jailbreaking.
>
> Due to time constraints, we conducted the performance evaluation against jailbreaking examples only for the LLaMA-7B model.. The results, summarized in the table below, demonstrate that our method significantly outperforms STL in both Attack Success Rate, ASR (lower is better) and Mean Judge Score (lower is better). Notably, our method achieves a 19.57% ASR and a 6.02 Mean Judge Score, indicating superior robustness. Additionally, the robustness improves as the number of unsafe examples used during training increases.
>
> | method | Finetuning data | ASR | Mean Judge Score |
> | :---- | :---- | :---- | :---- |
> | Ours | Alpaca+1000 unsafe examples | **19.57%** | **6.02** |
> | Ours | Alpaca+500 unsafe examples | 26.09% | 7.39 |
> | Ours | Alpaca+300 unsafe examples | 34.78% | 8.11 |
> | Ours | Alpaca+100 unsafe examples | 63.04% | 9.13 |
> | STL | Alpaca+1000 safe examples | 60.00% | 8.91 |
> | STL | Alpaca+500 safe examples | 63.04% | 8.71 |
> | STL | Alpaca+300 safe examples | 73.33% | 9.33 |
> | STL | Alpaca+100 safe examples | 69.57% | 9.07 |
>
> # Experiments With Various Sample Sizes for Over Alignment Issue.
>
>  We sincerely appreciate the reviewer’s recognition that the overalignment issue identified in the paper is a valuable starting point. To further evaluate this issue, we conducted additional experiments using LLaMA-7B trained with our TA-SFT approach, leveraging 1000, 500, 300, and 100 unsafe examples. We also examined LLaMA-7B models trained with Safety-Tuned-LLaMA (STL) using 10k, 4k, 2k, 1k, and 300 safe examples. However, consistent with the findings presented in Figure 7 of the main paper, we observed that the Toxic-Rejection-Rate versus Seemingly-Toxic-Rejection-Rate results remain on the same curve. Based on these findings, we conclude that **the number of safe or unsafe examples used in training does not explain the over alignment issue**.
>
> While the over alignment issue is a remaining noteworthy challenge, solving it is not the primary focus of this paper. Identifying the root cause of overalignment remains an exciting avenue for future research.
>
> *We will update the PDF with our new results and typo fixes after performing all experiments asked by all reviewers.*

---

> > ### Comment · Reviewer_L943 · 2024-11-20
> >
> > Thanks for the experiments!
> >
> > The Jailbreak experiments look really positive and confirm the robustness of the training. Updating my score to 6.
> >
> > I think I should have better expressed my second point. I wanted to understand if unsafe answers are ending up to be useful because we are training on actual content. In the safe answers case, we only train on a refusal answer so there is not much to be learnt. Hence I wanted to see if a mixture of safe and unsafe ends up useful as well.
> > A more rigorous (and time taking?) approach would be to annotate/somehow obtain actual safe answers to the unsafe questions and train on them.

---

> > > ### Author Response · Authors · 2024-11-20
> > >
> > > Thank you for your thoughtful feedback and for raising your score. We deeply appreciate your clarification of the second point and your valuable insight that "training on a refusal answer leaves little to be learned."
> > >
> > > The idea of using actual safe answers—beyond simple refusals like "Sorry, I cannot..."—to train LLMs is both intriguing and promising. As a preliminary step, we plan to experiment with this approach and hopefully complete it within the rebuttal period.
> > >
> > > What we have noted is that GPT-3.5 Turbo struggled to generate meaningful safe responses to toxic prompts, but we observed that the latest GPT-4o model performs better, offering more nuanced and constructive answers. Thus, we plan to experiment further by leveraging advanced LLMs to generate safe responses. We will apply rigorous filtering to exclude simple refusals and explore iterative refinement processes to ensure the responses are both safe and informative.
> > >
> > > We will prioritize incorporating these approaches during the rebuttal phase. Thank you again for your insightful comments, which have greatly strengthened our work.

---

> ### Comment · Reviewer_L943 · 2024-11-20
>
> The proposed experiment sounds good. I might be nitpicking but I am not sure the comparison between (training with unsafe answers from LLaMA) and (training with safe answers from GPT4o) is completely fair. Nevertheless, in a perfectly ideal scenario we need answers with similar information content but one safe and other unsafe, which is hard and time-taking to design.
>
> The main question here is
>
> **Q) Which is better in safety training: (a) pushing towards safe answers and/or (b) pushing away from unsafe answers.**
>
> We could not have asked this question without this work, hence this is a great contribution to the community. I would love to look at any preliminary results you have by the end of rebuttal.

---

> ### Author Response · Authors · 2024-11-25
>
> Thank you for pointing out the fair comparison issue in our experimental plan. Your insightful comment has further refined our understanding and approach.
>
> While collecting human-annotated data is indeed the most ideal solution, it is very expensive and time-consuming. To address this, we still seek to utilize an existing dataset and identified the **PKU-SafeRLHF** dataset as particularly suitable for our purpose. This dataset, available on HuggingFace, includes toxic prompts. Each prompt is paired with two responses, accompanied by labels indicating which response is better and **whether it is safe**.
>
> To ensure the dataset meets our requirements for fair comparison, we applied the following filtering steps:
>
> 1. Keep entries where both safe and unsafe responses are present for the same prompt.
> 2. Exclude entries where the safe response begins with a direct rejection (e.g., "I'm sorry, I cannot..."), focusing on actual safe responses, where we use a dictionary of direct rejection responses from the prior work \[1\].
> 3. Segregate the data based on the source of the responses, as these are generated by Alpaca, Alpaca2-7B, and Alpaca3-8B. For simplicity, we refer to the cleaned subsets as safety\_1, safety\_2, and safety\_3, respectively.
>
> After cleaning, we set up three datasets with approximately 3,000 toxic prompts in each of them, each paired with an actual safe response and a corresponding unsafe response.
>
> Using these datasets, we plan to conduct a fair comparison by training LLaMA-7B with the same prompts and safe/unsafe responses from the same LLMs using two approaches:
>
> 1. **TA-SFT with EMD (ours)**: Trained using instruction-following data mixed with unsafe examples.
> 2. **STL**: Trained using instruction-following data mixed with safe examples.
>
> However, it is important to note a key difference between this experimental setup and the pipeline described in our main paper. In our main paper, unsafe responses were generated by the target LLM itself, whereas in this experiment, unsafe responses are sourced from external LLMs (Alpaca models). This difference may result in a performance decline when applying our TA-SFT method.
>
> To explore the effect of data size, we will train LLaMA-7B on a mix of safety-unrelated data and varying amounts of safety-related data sampled from safety\_1 and Safety\_3. Due to limited time, we don’t test on Safety\_2. Additionally, since we observed that oversampling safety-related data improves STL performance when training with safe responses (pointed out by the third Reviewer hhQn), we will include oversampling in our comparison for a more comprehensive analysis. In the ‘Finetuning Data’ column of the following table, ‘Alpaca+Safety\_1-xxx-y’ indicate that the oversampling rate of y times oversampling for the number of xxx safety-related data sampled from Safety\_1.
>
> We report the number of harmful responses on the four safety evaluation benchmarks used in our main paper. Surprisingly, we find that **training with actual safe responses using the STL method does not improve the safety level at al**l. While our method, trained with unsafe responses obtained from external LLMs (different from the dataset pipeline in the main paper), experiences a slight performance decrease, it still achieves a satisfying safety level. This finding suggests that our method is robust even when using datasets sourced from external LLMs.
>
> Due to limited time, we have not explored training with actual safe response and unsafe response simultaneously using our method. However, based on our preliminary findings, we believe that ***pushing away from unsafe answers*** **appears to be a more robust approach to improving safety.** The underlying reason why training with actual safe responses fails to improve safety remains unclear and needs further investigation. We consider this an important direction for future work.
>
> | Finetuning Data | Method | I-MaliciousInstructions | I-CoNa | I-Controversial | HarmfulQ |
> | :---- | :---- | :---- | :---- | :---- | :---- |
> | Alpaca+Safety\_1-1000 | STL | 6 | 96 | 9 | 10 |
> | Alpaca+Safety\_1-1000-4 | STL | 9 | 99 | 8 | 7 |
> | Alpaca+Safety\_1-2000 | STL | 8 | 99 | 8 | 9 |
> | Alpaca+Safety\_1-3000 | STL | 9 | 99 | 9 | 6 |
> | Alpaca+Safety\_1-1000 | **Ours** | **2** | **15** | **1** | **3** |
> | Alpaca+Safety\_3-1000 | STL | 10 | 90 | 6 | 7 |
> | Alpaca+Safety\_3-1000-4 | STL | 6 | 75 | 3 | 8 |
> | Alpaca+Safety\_3-2000 | STL | 8 | 94 | 6 | 9 |
> | Alpaca+Safety\_3-3000 | STL | 8 | 98 | 5 | 10 |
> | Alpaca+Safety\_3-1000 | **Ours** | **2** | **6** | **0** | **1** |
>
> \[1\] Röttger P, Kirk H R, Vidgen B, et al. Xstest: A test suite for identifying exaggerated safety behaviours in large language models\[J\]. arXiv preprint arXiv:2308.01263, 2023\.

---

> > ### Comment · Reviewer_L943 · 2024-11-28
> >
> > Thanks for these results. They will greatly enhance the comprehensiveness of the work. All my queries are resolved now.

---

> > > ### Author Response · Authors · 2024-11-28
> > >
> > > Thank you very much for your kind comment and for acknowledging the additional results. We sincerely appreciate your thoughtful engagement and valuable feedback throughout the discussion, which has greatly contributed to improving the comprehensiveness of our work.

---

### Author Response · Authors · 2024-11-26

We sincerely thank all reviewers for their constructive and insightful feedback, as well as for recognizing the novelty of our TA-SFT method  and the data efficiency of our approach. Your input has been instrumental in helping us improve our submission. In response to your comments, we have made the following major updates:

1. Enhanced Robustness Evaluation:
We conducted additional experiments to compare the robustness of our method (TA-SFT) with the baseline Safety-Tuned LLaMA (STL) under jailbreaking attacks. Our results demonstrate that TA-SFT exhibits superior robustness compared to STL.
2. Manuscript Refinements:
A comprehensive review of the manuscript was carried out to address formatting issues and improve clarity and expression throughout.
3. Computational Analysis:
We provided a theoretical analysis of the additional computational workload introduced by TA-SFT with the EMD term. Empirical results show that TA-SFT training time is only 1–2% longer than standard SFT, highlighting its scalability for large datasets.
4. Extended Dataset Training:
We trained LLaMA-7B on a dataset 2.5 times larger than the one used in the main paper. The results indicate that our method maintains comparable response quality while improving safety levels. This suggests that TA-SFT has the potential for further performance gains when trained on even larger datasets.

Additionally, we try to address the question posed by Reviewer L943: "Which is better in safety training: (a) pushing towards safe answers and/or (b) pushing away from unsafe answers?" Our preliminary findings indicate that pushing towards actual safe answers (beyond simple rejections like "Sorry, I cannot...") does not enhance safety levels at all. However, our method, trained on data sourced from external LLMs, consistently performs well. These results suggest that **pushing away from unsafe answers is a more robust strategy for improving safety**.

These analyses, its results, and answers to further reviewer questions are contained in our responses to the individual reviews below. Corresponding figures and tables are contained in the Appendix. We again want to thank the reviewers for their valuable time, attention and for actively taking part in the review process.

---

### Meta-Review · Area_Chair_nYHu · 2024-12-23

**Metareview:**

This is a LLM safety paper that tackles self-generating safety data. The motivation is to overcome the need for large amounts of expensive human-generated data. Concretely, the authors seek to enhance safety at the SFT stage with limited amounts of safety-oriented human data. The way this is done is by using the small amounts of safety data, where a new term in the loss pushes next token prediction away from toxic outputs. This pushing is done via a practical proxy to the Wasserstein/EMD distance.

The paper has a number of strengths, including a simple idea, solid results in a challenging setting, and a good set of hypotheses studied, with some interesting insights generated. The latter is particularly interesting when discussing the ‘overalignment’ issue

For weaknesses, the authors are effectively doing a type of alignment during SFT—this is the implication of the loss adjustment on the particular dataset. This is fine, but then we’d expect a comparison with alignment methods (RLHF, DPO, etc), which the authors explicitly argue they should not have to compare with.

The latter weakness is not absolutely critical, though it would be nice for the authors to comment on this for camera ready. Overall this was a nice paper and worth accepting.

**Additional Comments On Reviewer Discussion:**

The reviewers brought up a number of good points during rebuttal (e.g., the relationship with jailbreaking) along with some additional experimental requests. The authors addressed basically everything here.

---

### Decision · Program_Chairs · 2025-01-22

Accept (Poster)